# FairerCLIP: Debiasing CLIP's Zero-Shot Predictions using Functions in RKHSs

**Sepehr Dehdashtian**[*]    **Lan Wang**[*]    **Vishnu Naresh Boddeti**
Michigan State University
{sepehr, wanglan3, vishnu}@msu.edu

## Abstract

Large pre-trained vision-language models such as CLIP provide compact and general-purpose representations of text and images that are demonstrably effective across multiple downstream zero-shot prediction tasks. However, owing to the nature of their training process, these models have the potential to 1) propagate or amplify societal biases in the training data and 2) learn to rely on spurious features. This paper proposes FairerCLIP, a general approach for making zero-shot predictions of CLIP more fair and robust to spurious correlations. We formulate the problem of jointly debiasing CLIP's image and text representations in reproducing kernel Hilbert spaces (RKHSs), which affords multiple benefits: 1) *Flexibility:* Unlike existing approaches, which are specialized to either learn with or without ground-truth labels, FairerCLIP is adaptable to learning in both scenarios. 2) *Ease of Optimization:* FairerCLIP lends itself to an iterative optimization involving closed-form solvers, which leads to $4\times$-$10\times$ faster training than the existing methods. 3) *Sample Efficiency:* Under sample-limited conditions, FairerCLIP significantly outperforms baselines when they fail entirely. And, 4) *Performance:* Empirically, FairerCLIP achieves appreciable accuracy gains on benchmark fairness and spurious correlation datasets over their respective baselines.

## 1 Introduction

Vision-Language Models such as CLIP (Radford et al., 2021) are trained on large-scale datasets of image-text pairs to learn representations that have high similarity for related image-text pairs. While these models have gained significant attention in recent years due to their remarkable zero-shot classification capabilities, they are not flawless. There is growing evidence that such models suffer from biases w.r.t. demographic (e.g., sex or skin tone) attributes (Agarwal et al., 2021; Wang et al., 2021; Birhane et al., 2023a;b; Dehdashtian et al., 2024) and even non-demographic(e.g., image background or illumination) attributes (Du et al., 2022; Zhang & Ré, 2022).

The above-mentioned biases can be viewed based on dependencies between the data attributes. We show these dependencies in Fig. 1: $X$ is the data (e.g., face images) that depends on some attributes, including $Y$, the target attribute we wish to predict, and $S$, the attribute that leads to bias. The goal of bias mitigation is to ensure that the

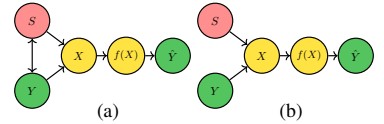

Figure 1: Dependence graphs for debiasing.

prediction $\hat{Y}$ is independent of $S$. We group the biases into those arising from two scenarios: (1) $Y$ and $S$ are dependent (Fig. 1 a): for example, *high cheekbones* as $Y$ and *sex* as $S$ since males typically have higher cheekbones than females. We refer to this type of correlation as **intrinsic dependence**. (2) $Y$ and $S$ are independent (Fig. 1 b): for example *hair color* as $Y$ and *sex* as $S$ since the hair color of a person does not depend on their sex. In this case, we refer to any observed correlation as a **spurious correlation**.

Several efforts (Zhang & Ré, 2022; Gao et al., 2021; Kumar et al., 2022; Kirichenko et al., 2022; Chuang et al., 2023; Wortsman et al., 2022; An et al., 2023; Adila et al., 2023), have been made to debias zero-shot predictions from CLIP models. However, they are limited in either one or more respects: (1) **Type of Bias:** Existing CLIP debiasing methods only consider spurious correlations

---

[*]Equal Contribution.

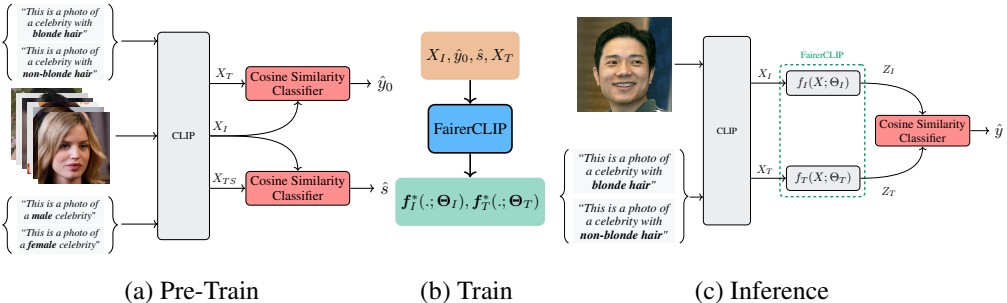

(a) Pre-Train  (b) Train  (c) Inference

Figure 2: Overview of the train and inference phases of FairerCLIP. (a) Shows the label prediction step. When labels are not available for training, FairerCLIP uses cosine similarity between the $X_T$ and $X_I$, and $X_{TS}$ and $X_I$ to predict the target attributes and sensitive attributes, respectively. (b) Shows the inputs and outputs for FairerCLIP in its training stage. FairerCLIP uses representation of images and the corresponding text prompts that are constructed by target attribute $(Y)$ along with the predicted labels to find the image and text encoders, i.e., $\boldsymbol{f}_I^*(.; \boldsymbol{\Theta}_I)$ and $\boldsymbol{f}_T^*(.; \boldsymbol{\Theta}_T)$. (c) Shows the inference phase of FairerCLIP in which we use the trained image and text encoders to generate debiased representations from the ones generated by CLIP.

in the data (Fig. 1 b) and do not seek to address bias induced by pairs of attributes with intrinsic dependencies (Fig. 1 a), (2) **Labels for training:** All existing approaches are tailored to train/fine-tune either with (supervised) or without (unsupervised) ground-truth labels and, as such, cannot be employed in both scenarios, (3) **Efficiency:** Some approaches adopt iterative methods to debias the features. However, they are computationally expensive to *train*, i.e., slow to converge, leading to high training latency and many parameters in the debiasing modules, increasing model sizes even further.

We propose FairerCLIP to address the aforementioned limitations of existing debiasing approaches. FairerCLIP affords sufficient flexibility to mitigate bias arising from both spurious correlations and intrinsic dependencies and, in both scenarios, learn with or without ground-truth labels. FairerCLIP utilizes a non-parametric measure of statistical dependence that accounts for all linear and non-linear relations between the debiased representation and the sensitive attribute of interest. Our formulation lends itself to alternating optimization, with each update having a closed-form solution and, in comparison to baselines, enjoying fast training convergence and requiring fewer parameters to train. An overview of FairerCLIP in its train and inference phases along with how we integrate this transformation over the underlying CLIP model is shown in Fig. 2.

**Summary of Contributions:** (1) We demonstrate that a single general method can debias the image and text features from frozen CLIP backbones under different scenarios more effectively than those specialized for each scenario. The scenarios include accounting for both spurious correlations and intrinsic dependencies (Sec. 4.2), learning with and without ground-truth labels (Sec. 4.2), and learning from small and medium-sized datasets (App. A.5). (2) We demonstrate that kernel methods are particularly effective compared to shallow MLPs when operating on features and optimizing possibly competing objectives, as is the case for debiasing CLIP representations. They enjoy closed-form solutions that allow for significantly faster training, can scale to medium-sized datasets, and are more effective under limited training data (Sec. 4.3, App. A.4, and Tab. 3 (left)).

## 2 THE DEBIASING CLIP REPRESENTATIONS PROBLEM

**Notation:** Scalars are denoted by regular lower case letters, e.g. $r$, $\tau$. Deterministic vectors are denoted by boldface lowercase letters, e.g., $\boldsymbol{x}$, $\boldsymbol{s}$. We denote both scalar-valued and multi-dimensional Random Variables (RVs) by regular uppercase letters, e.g. $X$, $S$. Deterministic matrices are denoted by boldface uppercase letters, e.g. $\boldsymbol{H}$, $\boldsymbol{\Theta}$, and the entry at $i^{th}$ row, $j^{th}$ column of matrix $\boldsymbol{M}$ is denoted by $(\boldsymbol{M})_{ij}$ or $m_{ij}$. $\boldsymbol{I}_n$ or simply $\boldsymbol{I}$ denotes an $n \times n$ identity matrix, $\mathbf{1}_n$ or $\mathbf{1}$ and $\mathbf{0}_n$ or $\mathbf{0}$ are $n \times 1$ vector of ones and zeros, respectively. We denote the trace of any square matrix $\boldsymbol{K}$ by $\text{Tr}[\boldsymbol{K}]$. Finite or infinite sets are denoted by calligraphy letters, e.g., $\mathcal{H}$, $\mathcal{A}$.

**Problem Setup:** We assume that the joint RV $(X_I, X_T, Y, S)$ contains the pre-trained image features $X_I \in \mathbb{R}^d$, pre-trained text features of target attribute $X_T \in \mathbb{R}^d$, target attribute $Y \in \mathbb{R}^{d_Y}$, and

sensitive attribute $S \in \mathbb{R}^{d_S}$. Their joint distribution will be $\boldsymbol{p}_{X_I, X_T, Y, S}$. Furthermore, $Y$ and $S$ can also belong to any finite set, such as a categorical set.

Our aim is to debias $X_I$ and $X_T$ by generating representations, $Z_I = \boldsymbol{f}_I(X_I)$ and $Z_T = \boldsymbol{f}_T(X_T)$, with no or reduced *statistical dependence* on $S$. To measure this dependency, we need to employ a metric capable of capturing both linear and non-linear statistical dependencies.

**Choice of Dependence Measure:** We will adopt a simplified definition of the Hilbert-Schmidt Independence Criterion (HSIC) (Gretton et al., 2005) introduced by Sadeghi et al. (2022), defined as,

$$\text{Dep}(Z, S) := \sum_{j=1}^{r} \sum_{\beta \in \mathcal{U}_S} \mathbb{C}\text{ov}^2 \left( Z_j, \beta(S) \right), \tag{1}$$

where $\mathcal{U}_S$ is a countable orthonormal basis set for the separable universal RKHS $\mathcal{H}_S$, and $Z_j = f_j(X)$ for $f_j \in \mathcal{H}_X \forall j = 1, ..., r$. $\text{Dep}(Z, S)$ can be estimated (see Lemma 1 in Sadeghi et al. (2022)) empirically as,

$$\text{Dep}(Z, S) := \frac{1}{n^2} \left\| \boldsymbol{\Theta} \boldsymbol{K}_X \boldsymbol{H} \boldsymbol{L}_S \right\|_F^2, \tag{2}$$

where $n$ is the number of data samples, $\boldsymbol{K}_X \in \mathbb{R}^{n \times n}$ is the Gram matrix corresponding to $\mathcal{H}_X$, $\boldsymbol{\Theta}$ is the encoder parameter in $\boldsymbol{f}(X) = \boldsymbol{\Theta}[k_{X_1}, k_{X_2}, \cdots, k_{X_n}]^T$, $\boldsymbol{H} = \boldsymbol{I}_n - \frac{1}{n} \boldsymbol{1}_n \boldsymbol{1}_n^T$ is the centering matrix, and $\boldsymbol{L}_S$ is a full column-rank matrix corresponding to the Cholesky factorization of $K_S$, i.e., $\boldsymbol{L}_S \boldsymbol{L}_S^T = \boldsymbol{K}_S$. While HSIC and related measures like Maximum Mean Discrepancy (MMD) Gretton et al. (2012) have been employed by prior fairness approaches (Bahng et al., 2020; Quadrianto et al., 2019; Jung et al., 2021), the HSIC variation we use in Eq. (2) has several attractive properties (Sadeghi et al., 2022). This includes a convergence rate of $\mathcal{O}(n^{-1/2})$[1], a practical ability to capture all non-linear modes of dependencies when projecting from a high-dimensional representation to a low-dimensional representation, and, as we demonstrate next, analytical tractability.

In addition to the above-mentioned dependence metric, we also need another metric that can mimic the cosine similarity-based classification employed by CLIP. This metric will be used to make the representations of images and their corresponding text prompts align with each other to improve the accuracy of the predictions. As a result, we modify the definition of Dep metric in Eq. (1) and use a linear kernel as $\beta$ in Lemma 1.

**Lemma 1.** Let $\boldsymbol{K}_{X_I}, \boldsymbol{K}_{X_T} \in \mathbb{R}^{n \times n}$ be the Gram matrices corresponding to $\mathcal{H}_{X_I}$ and $\mathcal{H}_{X_T}$, respectively, i.e., $(\boldsymbol{K}_{X_I})_{ij} = k_{X_I}(\boldsymbol{x}_{I_i}, \boldsymbol{x}_{I_j})$ and $(\boldsymbol{K}_{X_T})_{ij} = k_{X_T}(\boldsymbol{x}_{T_i}, \boldsymbol{x}_{T_j})$, where covariance is empirically estimated as

$$\mathbb{C}\text{ov}\left( f_j(X_I), g_m(X_T) \right) \approx \frac{1}{n} \sum_{i=1}^{n} f_j(\boldsymbol{x}_{I_i}) g_m(\boldsymbol{x}_{T_i}) - \frac{1}{n^2} \sum_{p=1}^{n} f_j(\boldsymbol{x}_{I_p}) \sum_{k=1}^{n} g_m(\boldsymbol{x}_{T_k}).$$

It follows that, the corresponding empirical estimator for $\text{Dep}(Z_I, Z_T)$ is

$$\text{Dep}(Z_I, Z_T) \quad = \quad \frac{1}{n^2} \left\| \boldsymbol{\Theta}_I \boldsymbol{K}_{X_I} \boldsymbol{H} \boldsymbol{K}_{X_T} \boldsymbol{\Theta}_T^T \right\|_F^2, \tag{3}$$

where $\boldsymbol{\Theta}_I$ and $\boldsymbol{\Theta}_T$ are the parameters of the image and text encoders, respectively, and $\boldsymbol{K}_{X_I}$ and $\boldsymbol{K}_{X_T}$ are the kernel matrices for the image and text features, respectively.

*Proof.* The main idea for proving equality equation 3 is to employ the representer theorem to express $f_j$ and $g_m$. The complete proof is available in the supplementary material. □

**Objective Function:** After choosing the appropriate dependence measure, we now define our objective function. Our goal is to mitigate bias in CLIP's zero-shot predictions by debiasing the underlying representations. This can be achieved by (1) reducing the information related to the sensitive attribute while (2) preserving information about the target attribute as much as possible in the pair of image-text representations and (3) keeping the image and corresponding text representations aligned with each other.

We formulate the above-mentioned learning objective through the following optimization problem.

---

[1]In scenarios where only a single or few samples are available, to an extent, heavy data augmentation can compensate for the lack of sufficient samples to accurately estimate Dep. However, this is beyond the scope of this paper, and all our experiments are performed without data augmentation.

**Definition 1.**

$$\sup_{\boldsymbol{f}_I \in \mathcal{A}_r^I, \boldsymbol{f}_T \in \mathcal{A}_r^T} \big\{ J\left(\boldsymbol{f}_I, \boldsymbol{f}_T, \tau_I, \tau_T, \tau_z\right) := \mathrm{Dep}\left(Z_I, Y\right) - \tau_I \mathrm{Dep}\left(Z_I, S\right)$$

$$+ \mathrm{Dep}\left(Z_T, Y\right) - \tau_T \mathrm{Dep}\left(Z_T, S\right) \tag{4}$$

$$+ \tau_z \mathrm{Dep}\left(Z_I, Z_T\right) \big\}$$

where $\mathrm{Dep}(\cdot, \cdot) \geq 0$ is the statistical dependence measure defined in Eq. (2). $\mathrm{Dep}(Q, U) = 0$ implies $Q$ is independent of $U$ (i.e., $Q \perp\!\!\!\perp U$), and $\mathrm{Dep}(Q, U) > 0$ implies $Q$ is dependent on $U$ (i.e., $Q \not\!\perp\!\!\!\perp U$), with larger values indicating greater degrees of dependence. $\tau_I$ and $\tau_T$ control the contribution of the corresponding debiasing terms and $\tau_z$ controls the alignment of the debiased image and text features $Z_I = \boldsymbol{f}_I(X_I)$ and $Z_T = \boldsymbol{f}_T(X_Y)$, respectively.

In the above definition, the terms $\mathrm{Dep}(Z_I, Y)$ and $\mathrm{Dep}(Z_T, Y)$ contribute to maximizing the statistical dependence between the representations and the target label $Y$, the terms $-\tau_I \mathrm{Dep}(Z_I, S)$ and $-\tau_T \mathrm{Dep}(Z_T, S)$ seek to make the representations independent of $S$, and the term $\tau_z \mathrm{Dep}(Z_I, Z_T)$ ensures that the text and image features are still aligned with each other after debiasing.

**Choice of Encoder:** We construct the mappings through a set of $r$ functions from $\mathbb{R}^{d_X} \to \mathbb{R}$ in a reproducing kernel Hilbert space (RKHS) $(\mathcal{H}_X, k_X(\cdot, \cdot))$, such as the RBF Gaussian kernel. Hence, the representation $Z$ can be expressed as,

$$Z = \boldsymbol{f}(X) := [Z_1, \cdots, Z_r]^T \in \mathbb{R}^r, \quad Z_j = f_j(X), f_j \in \mathcal{H}_X \; \forall j = 1, \ldots, r, \tag{5}$$

where $r$ is the dimensionality of the transformed representation.

Our choice of RKHS is motivated by several reasons. As we observe in Eq. (4), debiasing is inherently an optimization problem with multiple competing objectives. In such cases, optimization is the primary bottleneck rather than model expressivity. This was also observed in Sadeghi et al. (2022). The closed-form solution afforded by our approach mitigates the optimization challenges (Sec. 4.3 and App. A.6). RKHS has nice universal approximation properties and has performance comparable to shallow MLPs while being more computationally efficient for training (Sec. 4.3) and is performant under limited data scenarios (Sec. 4.2).

## 3 A SOLUTION TO THE DEBIASING CLIP REPRESENTATIONS PROBLEM

Given the choice of dependence measure in Eq. (2), the optimization problem in Eq. (4) can be expressed as,

$$\max_{\boldsymbol{f}_I \in A_r, \boldsymbol{f}_T \in A_r} \Big\{ J\left(\boldsymbol{f}_I, \boldsymbol{f}_T, \tau_I, \tau_T, \tau_z, \boldsymbol{X}_I, \boldsymbol{X}_T, \boldsymbol{Y}, \boldsymbol{S}\right) := \frac{1}{n^2} \left\| \boldsymbol{\Theta}_I \boldsymbol{K}_{X_I} \boldsymbol{H} \boldsymbol{L}_Y \right\|_F^2 - \tau_I \frac{1}{n^2} \left\| \boldsymbol{\Theta}_I \boldsymbol{K}_{X_I} \boldsymbol{H} \boldsymbol{L}_S \right\|_F^2$$

$$\frac{1}{n^2} \left\| \boldsymbol{\Theta}_T \boldsymbol{K}_{X_T} \boldsymbol{H} \boldsymbol{L}_Y \right\|_F^2 - \tau_T \frac{1}{n^2} \left\| \boldsymbol{\Theta}_T \boldsymbol{K}_{X_T} \boldsymbol{H} \boldsymbol{L}_S \right\|_F^2$$

$$+ \tau_z \frac{1}{n^2} \left\| \boldsymbol{\Theta}_I \boldsymbol{K}_{X_I} \boldsymbol{H} \boldsymbol{K}_{X_T} \boldsymbol{\Theta}_T^T \right\|_F^2 \Big\} \tag{6}$$

Our solution to the constrained optimization problem in Eq. (6) is based on the observation that it has a closed-form solution when either $\boldsymbol{f}_I$ or $\boldsymbol{f}_T$ are fixed. Let $\boldsymbol{Z}_O$ be the feature corresponding to the fixed parameter and $\boldsymbol{f}$ the optimization parameter of the other feature of interest. Then Eq. (6) reduces to two optimization problems of the following general form,

$$\max_{\boldsymbol{f} \in A_r} \big\{ J\left(\boldsymbol{f}, \tau, \tau_z, \boldsymbol{X}, \boldsymbol{Y}, \boldsymbol{S}, \boldsymbol{Z}_O\right) := \frac{1}{n^2} \left\| \boldsymbol{\Theta} \boldsymbol{K}_X \boldsymbol{H} \boldsymbol{L}_Y \right\|_F^2 - \tau \frac{1}{n^2} \left\| \boldsymbol{\Theta} \boldsymbol{K}_X \boldsymbol{H} \boldsymbol{L}_S \right\|_F^2 + \tau_z \frac{1}{n^2} \left\| \boldsymbol{\Theta} \boldsymbol{K}_X \boldsymbol{H} \boldsymbol{Z}_O \right\|_F^2 \big\} \tag{7}$$

This is easy to see since fixing either of the parameters in Eq. (6) renders the terms involving them to a constant w.r.t. the variable of interest, and hence can be ignored during optimization.

**Theorem 2.** Let the Cholesky factorization of $\boldsymbol{K}_X$ be $\boldsymbol{K}_X = \boldsymbol{L}_X \boldsymbol{L}_X^T$, where $\boldsymbol{L}_X \in \mathbb{R}^{n \times d}$ $(d \leq n)$ is a full column-rank matrix. Let $r \leq d$, then a solution to Eq. (7) is

$$\boldsymbol{f}^{\mathrm{opt}}(X) = \boldsymbol{\Theta}^{\mathrm{opt}} \left[ k_X(\boldsymbol{x}_1, X), \cdots, k_X(\boldsymbol{x}_n, X) \right]^T,$$

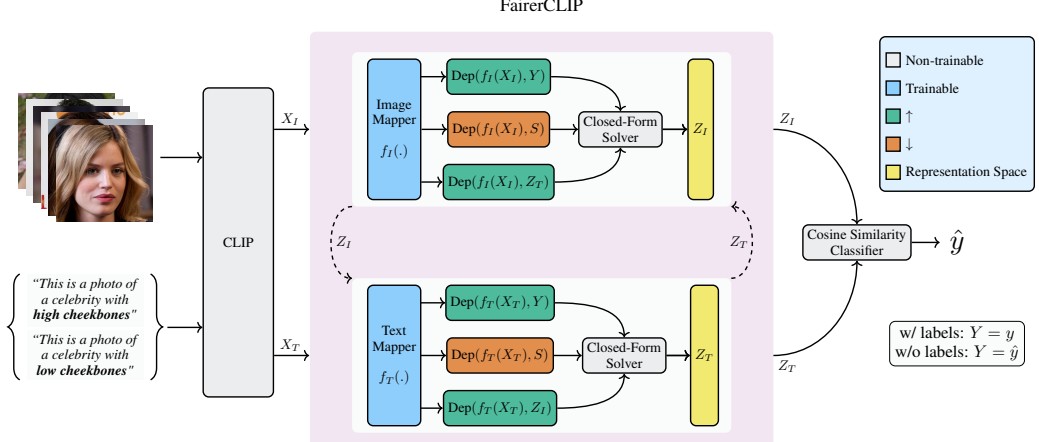

Figure 3: FairerCLIP acts on representations extracted from a frozen CLIP model. It has two mapping functions, $\boldsymbol{f}_I$ and $\boldsymbol{f}_T$, for the image and text representations. These functions are learned through an alternating optimization algorithm with two closed-form solvers. When ground-truth labels are unavailable for training, FairerCLIP learns from pseudo-labels $\hat{y}$, which are initialized from CLIP's zero-shot predictions and refined iteratively. The bold words in the input text prompts are the information of the target task label included in the text prompts.

where $\boldsymbol{\Theta}^{\text{opt}} = \boldsymbol{U}^T \boldsymbol{L}_X^{\dagger}$ and the columns of $\boldsymbol{U}$ are eigenvectors corresponding to the $r$ largest eigenvalues of the following generalized eigenvalue problem.

$$\boldsymbol{L}_X^T \left( \boldsymbol{H}\boldsymbol{K}_Y\boldsymbol{H} - \tau\boldsymbol{H}\boldsymbol{K}_S\boldsymbol{H} + \tau_z\boldsymbol{H}\boldsymbol{Z}_O\boldsymbol{Z}_O^T\boldsymbol{H} \right) \boldsymbol{L}_X\boldsymbol{u} = \lambda \left( \frac{1}{n}\boldsymbol{L}_X^T\boldsymbol{H}\boldsymbol{L}_X + \gamma\boldsymbol{I} \right) \boldsymbol{u}. \tag{8}$$

Furthermore, the objective value of (7) is equal to $\sum_{j=1}^r \lambda_j$, where $\{\lambda_1, \cdots, \lambda_r\}$ are $r$ largest eigenvalues of Theorem 2.

*Proof.* The objective in Eq. (7) can be expressed as a trace optimization problem, which reduces to a generalized eigenvalue problem (Kokiopoulou et al., 2011). See the supplementary material for detailed proof. $\qquad\square$

Building upon the above closed-form solution, we adopt alternating optimization to solve Eq. (6), by fixing $\boldsymbol{f}_I$ and solving for $\boldsymbol{f}_T$ and vice-versa (Fig. 3). The formulation in Eq. (4) requires labels of the downstream target task $Y$ and the sensitive labels $S$ to learn FairerCLIP's parameters. While such labels are readily available for supervised learning and partially available for semi-supervised learning (Jung et al., 2022; Chen et al., 2023), this is not the case for unsupervised learning. Therefore, in this case, we initialize the labels $Y$ and $S$ by the original zero-shot predictions $\hat{Y}$ and $\hat{S}$ from CLIP (Fig. 2 a). Then we refine $\hat{Y}$ by predicting it after every iterative update of $\boldsymbol{f}_I$ and $\boldsymbol{f}_T$. However, note that we do not update $\hat{\boldsymbol{S}}$ in the same way since our initial prediction of $\hat{S}$ has the most information about the label $\hat{S}$, but as we debias the representations in the subsequent iterations, we remove the information of $S$. Therefore, updated values of $\hat{S}$ will lead to inaccurate estimates of $\text{Dep}(Z, S)$ and affect the overall optimization. This procedure is detailed in Algorithm 1 of App. A.1.

## 3.1 A Geometric Illustration of FairerCLIP

A geometric illustration of the steps that FairerCLIP takes to debias the representations is shown in Fig. 4. In theory, the RBF kernels used in our encoder ($\phi_I(X)$ and $\phi_T(X)$) map the image and text features into an infinite-dimensional space, where the samples corresponding to different target attributes are linearly separable. In the infinite-dimensional space, the encoder that optimizes Eq. (4) for $\boldsymbol{\Theta}_I$ and $\boldsymbol{\Theta}_T$ by alternating between closed-form solvers and seeks a direction for mapping the image and text features that have low angular distance w.r.t. the direction of (1) $Y$ labels (small $\alpha_{IY}$ and $\alpha_{TY}$), (2) $S_{\perp}$ (small $\alpha_{IS_{\perp}}$ and $\alpha_{TS_{\perp}}$), and (3) the other representation (small $\alpha_{IT}$).

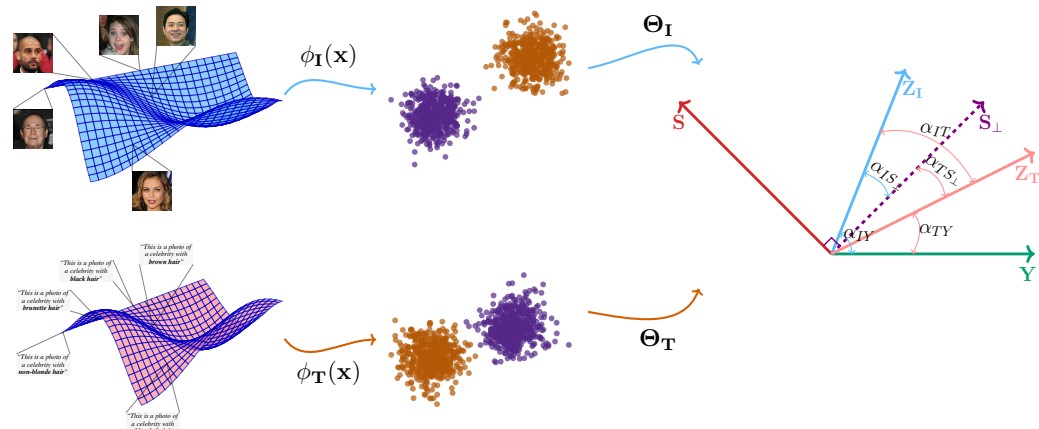

Figure 4: A geometric illustration of FairerCLIP training steps. The encoder utilizes the implicit mapping functions $\phi_I(X)$ and $\phi_T(X)$ of the RBF kernel to map image and text features into an infinite-dimensional space, facilitating linear separability of samples with different target attributes. The optimization process seeks a direction that aligns with labels $Y$, statistically independent of $S$, and aligned with the other representation.

# 4 EXPERIMENTAL EVALUATION

We evaluate FairerCLIP on datasets with spurious correlation and intrinsic dependence and compare it to several existing baselines. In summary, the experimental results indicate that the baseline methods are effective in mitigating spurious correlations, but they are not as effective at mitigating the bias caused by the intrinsic dependencies. In contrast, FairerCLIP effectively and efficiently mitigates both spurious correlations and intrinsic dependencies in CLIP's zero-shot predictions. In all our experiments, to overcome the $\mathcal{O}(n^3)$ computational and $\mathcal{O}(n^2)$ memory complexity of the kernel matrices $\boldsymbol{K}$, we use random Fourier features (RFF) (Rahimi & Recht, 2007). All the implementation details are provided in App. A.3

## 4.1 DATASETS

We evaluate FairerCLIP on an assortment of classification tasks across many datasets. This includes **Waterbirds** (Sagawa et al., 2019), which contains spurious correlations between the types of birds and background of the images, different settings of **CelebA** (Liu et al., 2015) that contains more than 200,000 face images of the celebrities in the wild annotated with 40 binary attributes and contains both spurious correlations and intrinsic dependencies among its attributes, **FairFace** dataset (Karkkainen & Joo, 2021) which contains more than 108,000 face images from 7 different race groups (White, Black, Indian, East Asian, Southeast Asian, Middle Eastern, and Latino) collected from the YFCC-100M Flickr dataset and labeled with race, sex, and age groups, and **Chicago Face Database (CFD)** (Ma et al., 2015) which includes face images with different annotations such as facial attributes, ethnicity, age, and sex.

## 4.2 EMPIRICAL RESULTS

We report the results of FairerCLIP and compare them with the performance of related baselines on various datasets and settings. Following the experimental settings of prior work (Zhang & Ré, 2022; Koh et al., 2021; Chuang et al., 2023), we do not presume sensitive attributes ($S$) during the training process but assume them in the validation dataset for hyperparameter tuning and model selection, as proposed in Koh et al. (2021). Thus, following prior work (Zhang & Ré, 2022), for FairerCLIP and other baselines that need label $S$, we use the zero-shot predictions of $S$ ($\hat{S}$) from CLIP as the sensitive attribute.

### 4.2.1 MITIGATING INTRINSIC DEPENDENCY

To evaluate the ability of FairerCLIP to mitigate intrinsic dependency, we conduct numerical evaluations on the CelebA dataset with *high cheekbones* as the target attribute and *sex* as the sensitive

attribute. As discussed in Sec. 1, these two attributes are intrinsically dependent. To measure the fairness of predictions, we employ the Equal Opportunity Difference (EOD) (Hardt et al., 2016) metric, defined as, $\text{EOD} := \left| P(\hat{Y} = 1 | Y = 1, S = 1) - P(\hat{Y} = 1 | Y = 1, S = 0) \right|$, where $S$ is the sensitive attribute, and $\hat{Y}$ and $Y$ are the predicted and the ground-truth target labels, respectively. Our choice of EOD is justified since other fairness definitions, like Demographic Parity Violation (DPV), are not well suited for many practical scenarios (Hardt et al., 2016; Chouldechova, 2017).

Table 1 compares the performance of FairerCLIP and the baselines on the CelebA dataset with intrinsic dependency. For this experiment, we train all methods except the zero-shot baseline, which is included to demonstrate the level of unfairness in the CLIP features with the ground-truth labels. We observe that among all baselines, Contrastive Adapter (Zhang & Ré, 2022) performs well and achieves appreciable EOD for the CLIP ResNet-50 model. However, most other methods seem to even amplify the bias in the original CLIP features while improving average accuracy. FairerCLIP performs the best in terms of debiasing, achieving an EOD of 0.002% and 0.005% for CLIP ResNet-50 and CLIP ViT-L/14, respectively. Overall, FairerCLIP is very effective at mitigating unfairness to a significant extent, achieving an EOD value close to zero while maintaining a high classification accuracy.

Table 1: Fairness on the CelebA dataset with intrinsic dependency. All values are in %.

| Method | CLIP ResNet-50 | | CLIP ViT-L/14 | |
|---|---|---|---|---|
| | Avg | EOD | Avg | EOD |
| Zero-shot (Radford et al., 2021) | 50.5 | 5.8 | 48.8 | 2.8 |
| ERM Linear Probe (Kumar et al., 2022) | 84.8 | 19.0 | 84.8 | 14.0 |
| ERM Adapter (Gao et al., 2021) | 85.3 | 11.0 | 84.6 | 14.0 |
| DFR (Subsample) (Kirichenko et al., 2022) | 83.2 | 4.2 | 84.1 | 7.4 |
| DFR (Upsample) (Kirichenko et al., 2022) | 83.6 | 4.1 | 84.1 | 6.6 |
| Contrastive Adapter (Zhang & Ré, 2022) | 84.2 | 1.0 | 83.6 | 6.3 |
| FairerCLIP (ours) | 83.4 | **0.02** | 83.8 | **0.005** |

### 4.2.2 Mitigating Spurious Correlation

To evaluate FairerCLIP's effectiveness in mitigating spurious correlation, we perform numerical experiments on spurious correlation benchmarks, Waterbirds and CelebA, following the settings in Zhang & Ré (2022). Since FairerCLIP can be learned with or without ground-truth labels, we compare it against methods from these two categories. For performance evaluation, we use three metrics: 1) Average accuracy (**Avg.**), 2) Worst-Group accuracy (**WG**), i.e., the lowest accuracy of all subgroups, and 3) **Gap**, which is the difference between average and worst-group accuracy.

Table 2: Comparison of prior methods and FairerCLIP in terms of worst group accuracy (WG), average accuracy (Avg), and their gap on the WaterBirds and CelebA datasets. For the latter, the target and sensitive attributes are blonde hair and sex for two different variants of CLIP, CLIP ResNet-50 and CLIP ViT-L/14, in two different settings–w/ and w/o labels. **1st** / 2nd best results are in **bold** / underlined.

| Method / Acc. | CLIP ViT-L/14 | | | | | | CLIP ResNet-50 | | | | | |
|---|---|---|---|---|---|---|---|---|---|---|---|---|
| | Waterbirds | | | CelebA | | | Waterbirds | | | CelebA | | |
| | WG (↑) | Avg (↑) | Gap (↓) | WG (↑) | Avg (↑) | Gap (↓) | WG (↑) | Avg (↑) | Gap (↓) | WG (↑) | Avg↑ | Gap (↓) |
| **w/ labels** | | | | | | | | | | | | |
| ERM Linear Probe (Kumar et al., 2022) | 65.4±0.5 | 97.7±0.1 | 32.3±0.5 | 30.4±1.5 | **94.6±0.1** | 64.2±1.5 | 13.2±0.7 | 94.6±0.1 | 81.4±0.7 | 13.1±0.9 | **94.8±0.0** | 81.6±0.8 |
| ERM Adapter (Gao et al., 2021) | 76.1±1.8 | **97.8±0.1** | 21.7±1.7 | 40.0±5.6 | 94.3±0.3 | 54.3±5.6 | 63.0±0.4 | **96.0±1.1** | 32.9±0.8 | 41.9±4.5 | 94.7±0.4 | 52.8±4.1 |
| DFR (Subsample) (Kirichenko et al., 2022) | 58.8±0.8 | 95.9±0.2 | 37.1±0.8 | 78.7±3.6 | 91.8±0.2 | 13.1±3.6 | 66.1±5.5 | 92.9±2.2 | 26.9±6.5 | 80.9±0.6 | 91.7±0.5 | 10.8±3.2 |
| DFR (Upsample) (Kirichenko et al., 2022) | 66.5±0.8 | 96.4±0.9 | 29.8±1.5 | 83.9±2.3 | 91.2±0.8 | 7.2±3.1 | 54.2±6.2 | 90.3±2.0 | 36.1±7.9 | **89.9±0.2** | 91.3±0.3 | **1.4±0.5** |
| Contrastive Adapter (Zhang & Ré, 2022) | 85.3±2.3 | 94.5±2.4 | 9.3±1.1 | 83.9±1.1 | 90.4±0.2 | 6.4±1.1 | **82.5±0.9** | 88.2±2.6 | **5.7±3.1** | 88.4±1.7 | 90.8±1.2 | 2.5±1.5 |
| FairerCLIP (ours) | **86.0±1.8** | 92.2±0.8 | **6.1±1.9** | **85.2±2.3** | 87.8±1.7 | **2.5±0.9** | 75.4±1.9 | 84.3±2.2 | 8.9±3.1 | 85±0.9 | 85±0.3 | 3.5±0.3 |
| **w/o labels** | | | | | | | | | | | | |
| Zero-shot (Radford et al., 2021) | 45.3±0.0 | 84.4±0.0 | 39.1±0.0 | 72.8±0.0 | 87.6±0.0 | 14.9±0.0 | 39.6±0.0 | 77.3±0.0 | 37.7±0.0 | 75.9±0.0 | 82.3±0.0 | 6.4±0.0 |
| Orth-Cali (Chuang et al., 2023) | 68.8 ± 0.0 | 84.5 ± 0.0 | 15.7±0.0 | 76.1±0.0 | 86.2±0.0 | 10.1±0.0 | 74.0±0.0 | 78.7±0.0 | 4.7±0.0 | 82.2±0.0 | 84.4±0.0 | 2.2±0.0 |
| FairerCLIP (ours) | **78.1±1.4** | **85.1±1.1** | **7.1±2.4** | **86.1±0.8** | **88.0±1.0** | **1.9±0.6** | **74.8±1.7** | **81.4±0.9** | **6.6±2.5** | **80.4±1.0** | **84.7±0.7** | **4.3±0.4** |

Tab. 2 shows the results of our empirical evaluation. We make the following observations: (i) On CLIP ViT-L/14, FairerCLIP has the lowest Gap and highest WG accuracy. (ii) For the CLIP ResNet-50, FairerCLIP outperforms the baselines in the w/o labels setting but not in the w/ label setting. The discrepancy between the performance FairerCLIP with CLIP ViT-L/14 and CLIP ResNet-50 can be attributed to the fact that CLIP ResNet-50 features contain less information about target attributes than CLIP ViT-L/14 features, as shown in App. A.7. Overall, the results of Tab. 2 indicate that FairerCLIP effectively improves the worst group's accuracy and reduces the Gap. Notably, our approach can be applied to and is effective in both scenarios, with and without ground-truth labels. At the same time, the baselines are specialized to operate in one or the other scenario only.

In Tab. 3, we evaluate FairerCLIP on the FairFace dataset. Here, we consider sex and race as the sensitive attributes and follow the experimental setup in Chuang et al. (2023). We use five target attributes and ten text prompts (2 prompts per attribute) unrelated to the samples' facial or sensitive attributes; we do not have access to ground-truth labels. As an example, the text prompt can be *"A photo of a **criminal** person"* or *"A photo of a **friendly** person"*. All the ten specific prompts are in App. A.3. To evalu-

ate the models, we calculate MaxSkew@1000 (Geyik et al., 2019), which assesses the maximum imbalance in certain sensitive attributes within a dataset. As is shown in Tab. 3 FairerCLIP outperforms the other baselines for both sensitive attributes across two different CLIP backbones.

In summary, the results in Tab. 2, Tab. 3, and Tab. 1 suggest that FairerCLIP can effectively mitigate the demographic bias caused by spurious correlation and intrinsic dependency in the data in both w/ and w/o the ground-truth labels settings.

Table 3: Comparison of FairerCLIP with baselines on FairFace dataset.

| Method / MaxSkew@1000 | CLIP ViT-B/32 | | CLIP ViT-L/14 | |
|---|---|---|---|---|
| | Sex | Race | Sex | Race |
| Zero-Shot (Radford et al., 2021) | 0.206 | 0.743 | 0.206 | 0.768 |
| Orth-Proj (Chuang et al., 2023) | 0.146 | 0.755 | 0.349 | 0.605 |
| Orth-Cali (Chuang et al., 2023) | 0.102 | 0.638 | 0.200 | 0.461 |
| FairerCLIP (ours) | **0.097** | **0.408** | **0.099** | **0.428** |

Next, we consider a more challenging task to evaluate the data-efficiency of FairerCLIP. We use CFD images with *attractive* and *sex* as the target and sensitive group attributes. The former is a continuous label, which we binarize by using the mean value of all samples as a threshold. Moreover, the sex attribute is a binary label. This task presents challenges in two aspects. First, the number of samples in this dataset is very small (597 samples), which may not be sufficient for training some of the baselines. Second, the performance of the zero-shot classifier for this case shows that the features generated by the CLIP model are not well separated, rendering it difficult to correctly predict $\hat{S}$ (see Appendix A.8). Fig. 5 shows the results of this experiment. We first observe that all the baselines almost completely fail at mitigating the bias for the worst group. In contrast, FairerCLIP's performance is satisfyingly better, both in terms of the worst group (WG) and average (Avg) accuracy. Furthermore, the Gap is significantly lower (21.8% vs 53.7% for (Zhang & Ré, 2022)).

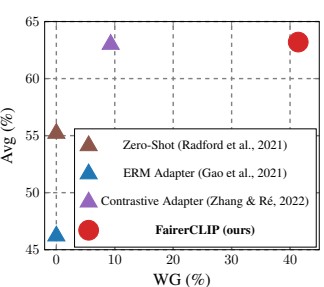

Figure 5: Results of FairerCLIP and baselines on CFD

### 4.3 COMPUTATIONAL EFFICIENCY OF TRAINING

To show the computational efficiency of FairerCLIP we report and compare the training time of FairerCLIP and other baselines in Tab. 4. The results show that FairerCLIP is an order of magnitude faster than most baselines and almost two orders faster than Contrastive Adapter (Zhang & Ré, 2022). The underlying model for this experiment is CLIP ViT-L/14, and all the numbers are measured on the same machine.

Table 4: Training time comparison (in seconds).

| Method | Waterbirds | CelebA |
|---|---|---|
| Contrastive Adapter(Zhang & Ré, 2022) | 1202 | 20602 |
| ERM Linear Probe(Kumar et al., 2022) | 157 | 2437 |
| ERM Adapter(Gao et al., 2021) | 161 | 1924 |
| DFR (Subsample)(Kirichenko et al., 2022) | 128 | 1878 |
| DFR (Upsample)(Kirichenko et al., 2022) | 176 | 2662 |
| FairerCLIP (ours) | **32** | **222** |

## 5 ABLATION STUDIES

We conduct systematic ablation studies in different settings to investigate the effectiveness of individual components of our approach. The settings include spurious correlation, intrinsic dependency experiments, and scenarios where ground-truth labels are unavailable. The results are shown in Tab. 5, where Tab. 5 (left) shows results of training w/ labels and Tab. 5 (right) shows training w/o labels. In the following, we describe each of these studies. For more ablation studies please refer to App. A.6.

**Effect of** $\text{Dep}(Z, Y)$ **term**: Here we study the effect of $\text{Dep}(Z, Y)$ by only retaining $\text{Dep}(Z, Y)$ in the objective. In this case, the worst group accuracy dropped by 14.5% for the group robustness experiment, and the EO increased to 0.195% for the fairness experiment. Although the new features were still able to maintain good separation w.r.t. $Y$, they lost their debiasing ability to a large extent.

**Effect of** $\text{Dep}(Z_I, Z_T)$ **term:** We remove $\text{Dep}(Z_I, Z_T)$ to investigate its effect on the alignment between the image and text embeddings. Results show that while maintaining a similar worst group accuracy, there is a decrease in average accuracy for both under the w/ and w/o ground-truth setups. This result demonstrates the contribution of this component in improving the predictions. Similarly, the results for the fairness experiment show that $\text{Dep}(Z_I, Z_T)$ also aids in enhancing the debiasing ability of FairerCLIP.

**Effect of updating** $\hat{Y}$**:** In this experiment, we predict $\hat{Y}$ once and fix it through the training process. Updating it during the training iterations has a considerable impact on worst group accuracy. The

Table 5: Ablation study w/ (left) and w/o (right) ground-truth labels on the CelebA dataset for different target attributes with sex as the sensitive attribute. We compare the effect of different components and parameters of FairerCLIP on its performance. Both $\text{Dep}(Z_I, Z_T)$ and $\text{Dep}(Z, Y)$ prove to be necessary and effective in maximizing the metrics. All values are in %.

| Method | Blonde Hair | | | High cheekbones | | Method | Blonde Hair | | |
| | WG | Avg | Gap | Avg | EOD | | WG | Avg | Gap |
|---|---|---|---|---|---|---|---|---|---|
| $\text{Dep}(Z, Y)$ only ($\tau = 0$) | 72.2 | 89.8 | 17.6 | 83.8 | 6.4 | $\text{Dep}(Z, Y)$ only ($\tau = 0$) | 75.0 | 87.7 | 12.7 |
| w/o $\text{Dep}(Z_I, Z_T)$ ($\tau_z = 0$) | 87.0 | 88.7 | 1.7 | 83.8 | 0.2 | w/o $\text{Dep}(Z_I, Z_T)$ ($\tau_z = 0$) | 81.8 | 86.1 | 4.3 |
| FairerCLIP | 86.7 | 89.3 | 2.6 | 83.8 | 0.005 | w/o updating $\hat{y}$ | 81.1 | 87.3 | 6.2 |
| | | | | | | FairerCLIP | 86.1 | 88.8 | 2.7 |

initial zero-shot accuracy for this group was $72.8\%$. Using the same initial $\hat{Y}$ during training improves the accuracy to $81.1\%$ while updating $\hat{Y}$ while training improves it further to $86.1\%$.

# 6 RELATED WORK

**CLIP and Bias:** Recent advancements in CLIP like models utilize multimodal data to learn representations that demonstrably generalize well to many downstream tasks and associated datasets (Radford et al., 2021; Desai & Johnson, 2021; Singh et al., 2022; Zellers et al., 2021; Zhang et al., 2021; Alayrac et al., 2022). Radford et al. (2021) demonstrated that utilizing a simple pretraining task with massive amounts of image-text pairs collected from the Internet, lead to models with strong transferability on different downstream tasks. FLAVA (Singh et al., 2022) learns representations by jointly pretraining on both unimodal and multimodal data. Flamingo Alayrac et al. (2022) demonstrated excellent generalization performance in few-shot and zero-shot scenarios. However, growing evidence (Birhane et al., 2023a;b) shows these models suffer from spurious correlations and bias towards certain demographic groups. For instance, Chuang et al. (2023) showed that textual prompt embeddings capture spurious correlations. In addition, Agarwal et al. (2021) discovered that zero-shot prediction from CLIP representations showed a high misclassification rate for certain races. Similarly, Wolfe et al. (2022) observed that CLIP embeddings exhibit stereotypes about sex and race. Dehdashtian et al. (2024) numerically characterize two near-optimal accuracy-fairness trade-offs and evaluate how far CLIP models are from them. Complementing these observations, we observe that CLIP exhibits high levels of demographic bias on the CFD and CelebA datasets.

**Debiasing CLIP:** Several approaches have been proposed to debias CLIP embeddings. Wang et al. (2021) addressed bias in image search by combining balanced sampling and pruning spuriously correlated embeddings. Wang et al. (2022) proposed a two-stage method that used learnable word vector prefixes and a re-representation Matrix for debiasing image retrieval problems. Berg et al. (2022) jointly trained an adversarial classifier and image-text contrastive loss, effectively reducing different bias measures. Zhang & Ré (2022) employed a contrastive adapter training strategy to enhance group robustness. Following the group robustness evaluation, Chuang et al. (2023) proposed to remove bias from text embeddings by projecting out the biased direction with text data only. Seth et al. (2023) adapted an additive residual learner module that separates the protected attribute information from the image representation generated by the visual encoder of CLIP.

# 7 CONCLUDING REMARKS

This paper proposed FairerCLIP to mitigate bias in zero-shot predictions from CLIP. It is versatile enough to mitigate bias caused by both spurious correlations and intrinsic dependencies in data and can be trained with or without ground-truth labels. Our key idea was to model the CLIP debiasing problem in reproducing kernel Hilbert spaces and employ a non-parametric statistical dependence measure that considers all linear and non-linear relations between the representation and the attribute of interest. Our solution in the form of an alternating optimization algorithm is effective across a diverse set of datasets, including Waterbirds, CelebA, FairFace, and the Chicago Face Database, spanning a variety of intrinsic dependencies and spurious correlations among attributes. Lastly, kernel-based approaches are underrepresented in current learning solutions, and FairerCLIP shows their strong potential for the type of task considered in this paper due to its flexibility, ease of optimization, and promising performance.

**Acknowledgements:** This work was supported in part by the National Science Foundation (award #2147116) and the Office of Naval Research (award #N00014-23-1-2417).

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

# A   APPENDIX

In our main paper, we proposed FairerCLIP to debias the text and image features from pre-trained vision-language models. Here, we provide some additional analysis to support our main results. The supplementary material is structured as follows:

1. Representation Disentanglement and Training Algorithm in App. A.1
2. Proofs Lemmas and Theorems in App. A.2
3. Implementation details in App. A.3
4. Analysis of analytical computational complexity and memory complexity in App. A.4
5. Effect of data size on the performance of FairerCLIP in App. A.5
6. More ablation studies in App. A.6
7. Comparison of CLIP ViT-L/14 and CLIP ResNet-50 in App. A.7
8. Comparison of more than 100 Zero-Shot CLIP models on CFD in App. A.8

## A.1   TECHNIQUAL DETAILS

**Representation Disentanglement:** To ensure that the mapping functions avoid learning representations with redundant information where different dimensions are highly correlated with each other, we seek a compact (Bengio et al., 2013) debiased embedding space. Therefore, we impose additional constraints on the representation. Specifically, we constrain the search space of the mapping functions $\boldsymbol{f}(\cdot)$ to learn a disentangled representation (Bengio et al., 2013) as follows

$$\mathcal{A}_r := \left\{ (f_1, \cdots, f_r) \mid f_i, f_j \in \mathcal{H}_X, \mathbb{C}\text{ov}\left(f_i(X), f_j(X)\right) + \gamma \langle f_i, f_j \rangle_{\mathcal{H}_X} = \delta_{i,j} \right\}. \tag{9}$$

In the above set, $\mathbb{C}\text{ov}\left(f_i(X), f_j(X)\right)$ part enforces the covariance matrix of $Z = \boldsymbol{f}(X)$ to be an identity matrix and encourages the variance of each entry of $Z$ to be one and different entries of $Z$ to be uncorrelated with each other. The regularization part, $\gamma \langle f_i, f_j \rangle_{\mathcal{H}_X}$ encourages the components of the mapping functions to be of unit norm and as orthogonal as possible to each other. These constraints also aid with numerical stability during empirical estimation (Fukumizu et al., 2007).

**Training Algorithm:** Details of training FairerCLIP are presented in Algorithm 1. FairerCLIP uses representation of images, representation of texts corresponding to target attribute labels ($X_T$), and representation of text corresponding to sensitive attribute labels ($X_{TS}$) as its inputs. FairerCLIP's goal is to find the image encoder ($\boldsymbol{f}_I^*$) and text encoder ($\boldsymbol{f}_T^*$) that can map the biased features generated by the CLIP to a debiased representation space. The training algorithm starts with initializing the label predictions. Since this algorithm is used for scenarios where we do not have access to the ground-truth labels of target attributes and sensitive attributes, we need to predict them by zero-shot classification from CLIP features. However, in scenarios where we have access to the true labels of the target attribute, we can skip the pseudo $Y$ initialization step and use the ground truth $Y$ instead. In the last step of initialization, we need to initialize the representation of images since we are using an alternating method to optimize the parameters of both the image encoder and text encoder. After the initialization step, we start to train both models alternatingly. After each optimization iteration, we update our prediction of $Y$ labels. After reaching the stop condition, the training process is complete.

## A.2   PROOFS

**Lemma 1.** Let $\boldsymbol{K}_{X_I}, \boldsymbol{K}_{X_T} \in \mathbb{R}^{n \times n}$ be the Gram matrices corresponding to $\mathcal{H}_{X_I}$ and $\mathcal{H}_{X_T}$, respectively, i.e., $(\boldsymbol{K}_{X_I})_{ij} = k_{X_I}(\boldsymbol{x}_{Ii}, \boldsymbol{x}_{Ij})$ and $(\boldsymbol{K}_{X_T})_{ij} = k_S(\boldsymbol{x}_{Ti}, \boldsymbol{x}_{Tj})$, where covariance is empirically estimated as

$$\mathbb{C}\text{ov}\left(f_j(X_I), g_m(X_T)\right) \approx \frac{1}{n} \sum_{i=1}^{n} f_j(\boldsymbol{x}_{I_i}) g_m(\boldsymbol{x}_{T_i}) - \frac{1}{n^2} \sum_{p=1}^{n} f_j(\boldsymbol{x}_{I_p}) \sum_{k=1}^{n} g_m(\boldsymbol{x}_{T_k}).$$

It follows that, the corresponding empirical estimator for $\text{Dep}\left(Z_I, Z_T\right)$ is

$$\text{Dep}\left(Z_I, Z_T\right) = \frac{1}{n^2} \left\| \boldsymbol{\Theta}_I \boldsymbol{K}_{X_I} \boldsymbol{H} \boldsymbol{K}_{X_T} \boldsymbol{\Theta}_T^T \right\|_F^2, \tag{10}$$

---

**Algorithm 1:** FairerCLIP Training Without Labels

---

**Input:** $\boldsymbol{X}_I \in \mathbb{R}^{n \times d}$, $\boldsymbol{X}_T \in \mathbb{R}^{|Y| \times d}$, $\boldsymbol{X}_{TS} \in \mathbb{R}^{|S| \times d}$, $m \in \mathbb{N}$
**Output:** $\boldsymbol{f}_I^*$, $\boldsymbol{f}_T^*$
**Initialize:**
$\hat{\boldsymbol{Y}}^{(0)} \leftarrow \left\{ \forall \boldsymbol{x}_I \in \boldsymbol{X}_I, \boldsymbol{x}_T \in \boldsymbol{X}_T \middle| \underset{\boldsymbol{x}_T}{\mathrm{argmax}} \frac{\boldsymbol{x}_I^T \boldsymbol{x}_T}{\|\boldsymbol{x}_I\|\|\boldsymbol{x}_T\|} \right\}$; /* initialize pseudo Y            */

$\hat{\boldsymbol{S}} \leftarrow \left\{ \forall \boldsymbol{x}_I \in \boldsymbol{X}_I, \boldsymbol{x}_{TS} \in \boldsymbol{X}_{TS} \middle| \underset{\boldsymbol{x}_{TS}}{\mathrm{argmax}} \frac{\boldsymbol{x}_I^T \boldsymbol{x}_{TS}}{\|\boldsymbol{x}_I\|\|\boldsymbol{x}_{TS}\|} \right\}$; /* initialize pseudo S            */

$\boldsymbol{Z}_I^{(0)} \leftarrow \{ \boldsymbol{f}_I^*(\boldsymbol{X}_I) | \boldsymbol{f}_I^* \leftarrow \underset{\boldsymbol{f}_I}{\mathrm{argmax}} \, J(\boldsymbol{f}_I, \tau_I, 0, \boldsymbol{X}_I, \hat{\boldsymbol{Y}}^{(0)}, \hat{\boldsymbol{S}}, \boldsymbol{0}) \}$; /* Eq. (7)            */

$i \leftarrow 0$;
**while** $i < m$ **do**
$\quad$ $\boldsymbol{Z}_T^{(i+1)} \leftarrow \{ \boldsymbol{f}_T^*(\boldsymbol{X}_T) | \boldsymbol{f}_T^* \leftarrow \underset{\boldsymbol{f}_T}{\mathrm{argmax}} \, J(\boldsymbol{f}_T, \tau_T, \tau_z, \boldsymbol{X}_T, \hat{\boldsymbol{Y}}^{(i)}, \hat{\boldsymbol{S}}, \boldsymbol{Z}_I^{(i)}) \}$; /* solve Eq. (7)

$\quad$ $\boldsymbol{Z}_I^{(i+1)} \leftarrow \{ \boldsymbol{f}_I^*(\boldsymbol{X}_I) | \boldsymbol{f}_I^* \leftarrow \underset{\boldsymbol{f}_I}{\mathrm{argmax}} \, J(\boldsymbol{f}_I, \tau_I, \tau_z, \boldsymbol{X}_I, \hat{\boldsymbol{Y}}^{(i)}, \hat{\boldsymbol{S}}, \boldsymbol{Z}_T^{(i)}) \}$; /* solve Eq. (7)   */

$\quad$ $\hat{\boldsymbol{Y}}^{(i+1)} \leftarrow \left\{ \forall \boldsymbol{z}_I \in \boldsymbol{Z}_I^{(i+1)}, \boldsymbol{z}_T \in \boldsymbol{Z}_T^{(i+1)} \middle| \underset{\boldsymbol{z}_T}{\mathrm{argmax}} \frac{\boldsymbol{z}_I^T \boldsymbol{z}_T}{\|\boldsymbol{z}_I\|\|\boldsymbol{z}_T\|} \right\}$; /* refine pseudo Y       */

$\quad$ $i \leftarrow i + 1$
**end**

---

*Proof.*

$$
\begin{aligned}
\mathrm{Dep}(Z_I, Z_T) & := \sum_{m=1}^{r} \sum_{j=1}^{r} \left\{ \frac{1}{n} \sum_{i=1}^{n} f_j(\boldsymbol{x}_{I_i}) g_m(\boldsymbol{x}_{T_i}) - \frac{1}{n^2} \sum_{p=1}^{n} f_j(\boldsymbol{x}_{I_p}) \sum_{k=1}^{n} g_m(\boldsymbol{x}_{T_k}) \right\}^2 \\
& = \sum_{m=1}^{r} \sum_{j=1}^{r} \left\{ \frac{1}{n} \boldsymbol{\theta}_{I_j}^T \boldsymbol{K}_{X_I} \boldsymbol{K}_{X_T} \boldsymbol{\theta}_{T_m} - \frac{1}{n^2} \boldsymbol{\theta}_{I_j}^T \boldsymbol{K}_{X_I} \boldsymbol{1}_n \boldsymbol{1}_n^T \boldsymbol{K}_{X_T} \boldsymbol{\theta}_{T_m} \right\}^2 \\
& = \sum_{m=1}^{r} \sum_{j=1}^{r} \left\{ \frac{1}{n} \boldsymbol{\theta}_{I_j}^T \boldsymbol{K}_{X_I} \boldsymbol{H} \boldsymbol{K}_{X_T} \boldsymbol{\theta}_{T_m} \right\}^2 \\
& = \sum_{m=1}^{r} \frac{1}{n^2} \left\| \boldsymbol{\Theta}_I \boldsymbol{K}_{X_I} \boldsymbol{H} \boldsymbol{K}_{X_T} \boldsymbol{\theta}_{T_m} \right\|_2^2 \\
& = \frac{1}{n^2} \left\| \boldsymbol{\Theta}_I \boldsymbol{K}_{X_I} \boldsymbol{H} \boldsymbol{K}_{X_T} \boldsymbol{\Theta}_T^T \right\|_F^2
\end{aligned}
\tag{11}
$$

$\square$

**Theorem 2.** Let the Cholesky factorization of $\boldsymbol{K}_X$ be $\boldsymbol{K}_X = \boldsymbol{L}_X \boldsymbol{L}_X^T$, where $\boldsymbol{L}_X \in \mathbb{R}^{n \times d}$ $(d \leq n)$ is a full column-rank matrix. Let $r \leq d$, then a solution to

$$
\max_{\boldsymbol{f} \in A_r} \{ J(\boldsymbol{f}, \tau, \tau_z, \boldsymbol{X}, \boldsymbol{Y}, \boldsymbol{S}, \boldsymbol{Z}_O) := \frac{1}{n^2} \|\boldsymbol{\Theta} \boldsymbol{K}_X \boldsymbol{H} \boldsymbol{L}_Y\|_F^2 - \tau \frac{1}{n^2} \|\boldsymbol{\Theta} \boldsymbol{K}_X \boldsymbol{H} \boldsymbol{L}_S\|_F^2 + \tau_z \frac{1}{n^2} \|\boldsymbol{\Theta} \boldsymbol{K}_X \boldsymbol{H} \boldsymbol{Z}_O\|_F^2 \}
\tag{12}
$$

is

$$
\boldsymbol{f}^{\mathrm{opt}}(X) = \boldsymbol{\Theta}^{\mathrm{opt}} \left[ k_X(\boldsymbol{x}_1, X), \cdots, k_X(\boldsymbol{x}_n, X) \right]^T
$$

where $\boldsymbol{\Theta}^{\mathrm{opt}} = \boldsymbol{U}^T \boldsymbol{L}_X^\dagger$ and the columns of $\boldsymbol{U}$ are eigenvectors corresponding to the $r$ largest eigenvalues of the following generalized eigenvalue problem.

$$
\boldsymbol{L}_X^T \left( \boldsymbol{H} \boldsymbol{K}_Y \boldsymbol{H} - \tau \boldsymbol{H} \boldsymbol{K}_S \boldsymbol{H} + \tau_z \boldsymbol{H} \boldsymbol{Z}_O \boldsymbol{Z}_O^T \boldsymbol{H} \right) \boldsymbol{L}_X \boldsymbol{u} = \lambda \left( \frac{1}{n} \boldsymbol{L}_X^T \boldsymbol{H} \boldsymbol{L}_X + \gamma \boldsymbol{I} \right) \boldsymbol{u}.
\tag{13}
$$

Furthermore, the supremum value of the objective function is equal to $\sum_{j=1}^{r} \lambda_j$, where $\{\lambda_1, \cdots, \lambda_r\}$ are $r$ largest eigenvalues of equation 13.

*Proof.* Using the representer theorem, the disentanglement property in

$$
\mathcal{A}_r := \left\{ (f_1, \cdots, f_r) \mid f_i, f_j \in \mathcal{H}_X, \mathbb{C}\mathrm{ov}\left(f_i(X), f_j(X)\right) + \gamma \langle f_i, f_j \rangle_{\mathcal{H}_X} = \delta_{i,j} \right\}.
\tag{14}
$$

can be expressed as

$$\mathbb{C}\text{ov}\left(f_i(X), f_j(X)\right) + \gamma \langle f_i, f_j \rangle_{\mathcal{H}_X}$$

$$= \frac{1}{n}\sum_{k=1}^n f_i(\boldsymbol{x}_k)f_j(\boldsymbol{x}_k) - \frac{1}{n^2}\sum_{k=1}^n f_i(\boldsymbol{x}_k)\sum_{m=1}^n f_j(\boldsymbol{x}_m) + \gamma \langle f_i, f_j \rangle_{\mathcal{H}_X}$$

$$= \frac{1}{n}\sum_{k=1}^n\sum_{t=1}^n \boldsymbol{K}_X(\boldsymbol{x}_k,\boldsymbol{x}_t)\theta_{it}\sum_{m=1}^n \boldsymbol{K}_X(\boldsymbol{x}_k,\boldsymbol{x}_m)\theta_{jm} - \frac{1}{n^2}\boldsymbol{\theta}_i^T \boldsymbol{K}_X \boldsymbol{1}_n \boldsymbol{1}_n^T \boldsymbol{K}_X \boldsymbol{\theta}_j + \gamma \langle f_i, f_j \rangle_{\mathcal{H}_X}$$

$$= \frac{1}{n}\left(\boldsymbol{K}_X\boldsymbol{\theta}_i\right)^T\left(\boldsymbol{K}_X\boldsymbol{\theta}_j\right) - \frac{1}{n^2}\boldsymbol{\theta}_i^T \boldsymbol{K}_X \boldsymbol{1}_n \boldsymbol{1}_n^T \boldsymbol{K}_X \boldsymbol{\theta}_j + \gamma \left\langle \sum_{k=1}^n \theta_{ik}k_X(\cdot,\boldsymbol{x}_k), \sum_{t=1}^n \theta_{it}k_X(\cdot,\boldsymbol{x}_t) \right\rangle_{\mathcal{H}_X}$$

$$= \frac{1}{n}\boldsymbol{\theta}_i^T \boldsymbol{K}_X \boldsymbol{H} \boldsymbol{K}_X \boldsymbol{\theta}_j + \gamma\,\boldsymbol{\theta}_i^T \boldsymbol{K}_X \boldsymbol{\theta}_j$$

$$= \frac{1}{n}\boldsymbol{\theta}_i^T \boldsymbol{L}_X \left(\boldsymbol{L}_X^T \boldsymbol{H} \boldsymbol{L}_X + n\gamma\,\boldsymbol{I}\right)\boldsymbol{L}_X^T \boldsymbol{\theta}_j$$

$$= \delta_{i,j}.$$

As a result, $\boldsymbol{f} \in \mathcal{A}_r$ is equivalent to

$$\boldsymbol{\Theta}\boldsymbol{L}_X \underbrace{\left(\frac{1}{n}\boldsymbol{L}_X^T \boldsymbol{H} \boldsymbol{L}_X + \gamma\boldsymbol{I}\right)}_{:=\boldsymbol{C}}\boldsymbol{L}_X^T \boldsymbol{\Theta}^T = \boldsymbol{I}_r,$$

where $\boldsymbol{\Theta} := \left[\boldsymbol{\theta}_1, \cdots, \boldsymbol{\theta}_r\right]^T \in \mathbb{R}^{r\times n}$.

Let $\boldsymbol{V} = \boldsymbol{L}_X^T\boldsymbol{\Theta}^T$ and consider the optimization problem in equation 7:

$$\sup_{\boldsymbol{f}\in\mathcal{A}_r} \frac{1}{n^2}\left\{\|\boldsymbol{\Theta}\boldsymbol{K}_X\boldsymbol{H}\boldsymbol{L}_Y\|_F^2 - \tau\|\boldsymbol{\Theta}\boldsymbol{K}_X\boldsymbol{H}\boldsymbol{L}_S\|_F^2 + \tau_z\|\boldsymbol{\Theta}\boldsymbol{K}_X\boldsymbol{H}\boldsymbol{Z}_O\|_F^2\right\}$$

$$= \sup_{\boldsymbol{f}\in\mathcal{A}_r} \frac{1}{n^2}\left\{\text{Tr}\left\{\boldsymbol{\Theta}\boldsymbol{K}_X\boldsymbol{H}\boldsymbol{K}_Y\boldsymbol{H}\boldsymbol{K}_X\boldsymbol{\Theta}^T\right\} - \tau\text{Tr}\left\{\boldsymbol{\Theta}\boldsymbol{K}_X\boldsymbol{H}\boldsymbol{K}_S\boldsymbol{H}\boldsymbol{K}_X\boldsymbol{\Theta}^T\right\} + \tau_z\text{Tr}\left\{\boldsymbol{\Theta}\boldsymbol{K}_X\boldsymbol{H}\boldsymbol{Z}_O\boldsymbol{Z}_O^T\boldsymbol{H}\boldsymbol{K}_X\boldsymbol{\Theta}^T\right\}\right\}$$

$$= \max_{\boldsymbol{V}^T\boldsymbol{C}\boldsymbol{V}=\boldsymbol{I}_r} \frac{1}{n^2}\text{Tr}\left\{\boldsymbol{\Theta}\boldsymbol{L}_X\boldsymbol{B}\boldsymbol{L}_X^T\boldsymbol{\Theta}^T\right\}$$

$$= \max_{\boldsymbol{V}^T\boldsymbol{C}\boldsymbol{V}=\boldsymbol{I}_r} \frac{1}{n^2}\text{Tr}\left\{\boldsymbol{V}^T\boldsymbol{B}\boldsymbol{V}\right\} \tag{15}$$

where

$$\boldsymbol{B} := \boldsymbol{L}_X^T\left(\boldsymbol{H}\boldsymbol{K}_Y\boldsymbol{H} - \tau\boldsymbol{H}\boldsymbol{K}_S\boldsymbol{H} + \tau_z\boldsymbol{H}\boldsymbol{Z}_O\boldsymbol{Z}_O^T\boldsymbol{H}\right)\boldsymbol{L}_X$$

It is shown in Kokiopoulou et al. (2011) that an[2] optimizer of (15) is any matrix $\boldsymbol{U}$ whose columns are eigenvectors corresponding to $r$ largest eigenvalues of generalized problem

$$\boldsymbol{B}\boldsymbol{u} = \tau\,\boldsymbol{C}\boldsymbol{u} \tag{16}$$

and the maximum value is the summation of $r$ largest eigenvalues. Once $\boldsymbol{U}$ is determined, then, any $\boldsymbol{\Theta}$ in which $\boldsymbol{L}_X^T\boldsymbol{\Theta}^T = \boldsymbol{U}$ is optimal $\boldsymbol{\Theta}$ (denoted by $\boldsymbol{\Theta}^{\text{opt}}$). Note that $\boldsymbol{\Theta}^{\text{opt}}$ is not unique and has a general form of

$$\boldsymbol{\Theta}^T = \left(\boldsymbol{L}_X^T\right)^\dagger\boldsymbol{U} + \boldsymbol{\Lambda}_0, \quad \mathcal{R}(\boldsymbol{\Lambda}_0) \subseteq \mathcal{N}\left(\boldsymbol{L}_X^T\right).$$

However, setting $\boldsymbol{\Lambda}_0$ to zero would lead to minimum norm for $\boldsymbol{\Theta}$. Therefore, we opt $\boldsymbol{\Theta}^{\text{opt}} = \boldsymbol{U}^T\boldsymbol{L}_X^\dagger$. □

## A.3 IMPLEMENTATION DETAILS

We conducted experiments on CelebA, Waterbirds, FairFace, and the Chicago Face Dataset (CFD). For CelebA and Waterbirds, we follow their official train/val/test splits and only use ground truth

---

[2]Optimal $\boldsymbol{V}$ is not unique.

labels from the val split for hyperparameter tuning. For CFD, since there is no official dataset split, we randomly split it with a ratio of 0.5/0.1/0.4 for train/val/test. Following the standard setting Zhang & Ré (2022), we use val split to decide the optimal $\tau$, $\tau_z$, and dimensionality of the random Fourier features (RFF). For CelebA, the optimal $\tau$, $\tau_z$, and RFF dimensions are 0.8, 0.5, and 8000. For Waterbirds, the optimal $\tau$, $\tau_z$, and RFF dimensions are 0.7, 0.7, and 3000. And for CFD, the optimal $\tau$, $\tau_z$, and RFF dimensions are 0.6, 0.3, and 1000. In the scenario where the group labels are not available, we follow the same setup as the scenario where the group labels of the val split are available. For the CelebA dataset, we also conduct a pre-sampling process on the training split to balance the number of each class from the predicted $\hat{y}$.

For the FairFace dataset, we use 10 text prompts that are unrelated to the facial attributes or the sensitive attributes of the samples. In this setting, the sensitive attribute is gender or race, and the text prompts are constructed as *"This is a photo of a [attribute] person"* where *[attribute]* can be one of the elements of the {good, evil, smart, dumb, attractive, unattractive, lawful, criminal, friendly, unfriendly} set.

For all the above-mentioned experiments under different settings, we set the representation dimensionality $r$ to $c - 1$ where $c$ is the number of classes of the downstream target task.

For clarity, we summarized all the above-mentioned implementation details in Tab. 6

Table 6: Implementation details of FairerCLIP for each dataset.

| Dataset | RFF Dim. | $r$ | $\tau$ | $\tau_Z$ | Train/Val/Test | Training samples |
|---|---|---|---|---|---|---|
| CelebA | 8000 | 1 | 0.8 | 0.5 | Official Splits | 162,770 |
| Waterbirds | 3000 | 1 | 0.7 | 0.7 | Official Splits | 4,795 |
| FairFace | 3000 | 1 | 0.8 | 0.8 | Official Splits | 86,744 |
| CFD | 1000 | 1 | 0.6 | 0.3 | 0.5/0.1/0.4 | 298 |

### A.4 NUMERICAL COMPLEXITY

**Computational Complexity:** If $L_X$ in equation 13 is provided in the training dataset, then the computational complexity of obtaining the optimal encoder is $\mathcal{O}(l^3)$, where $l \leq n$ is the numerical rank of the Gram matrix $K_X$. However, the dominating part of the computational complexity is due to the Cholesky factorization, $K_X = L_X L_X^T$, which is $\mathcal{O}(n^3)$. Using random Fourier features (RFF) (Rahimi & Recht, 2007), $k_X(x, x')$ can be approximated by $r_X(x)^T r_X(x')$, where $r_X(x) \in \mathbb{R}^d$. In this situation, the Cholesky factorization can be directly calculated as

$$L_X = \begin{bmatrix} r_X(x_1)^T \\ \vdots \\ r_X(x_n)^T \end{bmatrix} \in \mathbb{R}^{n \times d}. \tag{17}$$

As a result, the computational complexity of obtaining the optimal encoder becomes $\mathcal{O}(d^3)$, where the RFF dimension, $d$, can be significantly less than the sample size $n$ with negligible error on the approximation $k_X(x, x') \approx r_X(x)^T r_X(x')$.

**Memory Complexity:** The memory complexity of equation 13, if calculated naively, is $\mathcal{O}(n^2)$ since $K_Y$, $K_S$, and $Z_O Z_O^T$ are $n$ by $n$ matrices. However, using RFF together with Cholesky factorization $K_Y = L_Y L_Y^T$, $K_S = L_S L_S^T$, the left-hand side of equation 13 can be re-arranged as

$$\left(L_X^T \tilde{L}_Y\right)\left(\tilde{L}_Y^T L_X\right) - \tau \left(L_X^T \tilde{L}_S\right)\left(\tilde{L}_S^T L_X\right) + \tau_z \left(L_X^T \tilde{Z}_O\right)\left(\tilde{Z}_O^T L_X\right), \tag{18}$$

where $\tilde{Z}_O^T = H Z_O = Z_O - \frac{1}{n}\mathbf{1}_n(\mathbf{1}_n^T Z_O)$ and $\tilde{L}_Y^T = H L_Y = L_Y - \frac{1}{n}\mathbf{1}_n(\mathbf{1}_n^T L_Y)$; therefore, the required memory complexity is $\mathcal{O}(nd)$. Note that $\tilde{L}_S^T$ and $H L_X$ can be calculated similarly.

### A.5 EFFECT OF DATA SIZE ON THE PERFORMANCE OF FAIRERCLIP

To evaluate the effectiveness of the FairerCLIP under limited data samples condition we report the performance of FairerCLIP as the size of the training dataset is varied. In this experiment, we

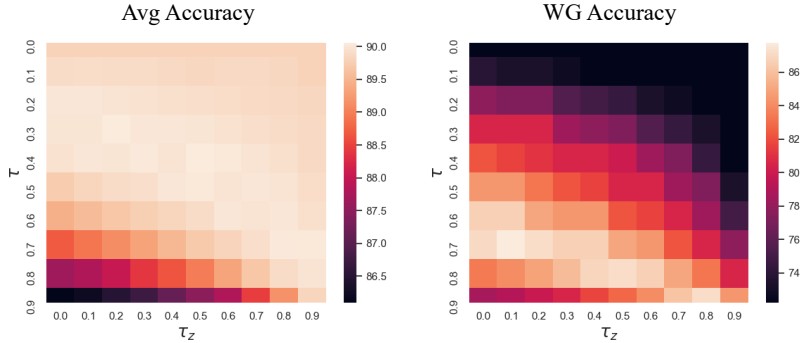

Figure 6: Effect of $\tau$ and $\tau_z$ on the average accuracy (left) and worst group accuracy (right). For every fixed $\tau_z$, the worst group accuracy increases with $\tau$ and then starts to decrease beyond a point. Additionally, the variation of the metrics is smooth over the range of hyperparameters, indicating lower sensitivity towards them.

randomly sampled 5, 25, 50, 75, and 100 percent of the training data as our training set. Then the Avg., WG, and Gap are evaluated. Table 7 shows the results of the evaluation for the Waterbird and CelebA datasets. The results indicate that FairerCLIP is able to perform sufficiently well when a small sub-sample of the dataset is used for its training. More specifically, in the CelebA dataset, FairerCLIP is only losing 2.9% of its WG accuracy when only 25% of the original training data is employed in its training phase. Moreover, on Waterbirds, a relatively smaller dataset, it loses less than 6% of WG accuracy when only 50% of training data is used. This experiment shows the effectiveness of FairerCLIP under a limited number of training samples.

Table 7: Effect of training data size on the performance of FairerCLIP

| # Samples | CelebA | | | Waterbird | | |
|---|---|---|---|---|---|---|
| | Avg. ($\uparrow$) | WG ($\uparrow$) | Gap ($\downarrow$) | Avg. ($\uparrow$) | WG ($\uparrow$) | Gap ($\downarrow$) |
| 5% | 84.21 | 73.88 | 10.32 | 86.31 | 65.26 | 21.05 |
| 25% | 88.26 | 83.88 | 4.37 | 86.54 | 78.66 | 7.88 |
| 50% | 86.65 | 84.29 | 2.36 | 89.11 | 81.31 | 7.80 |
| 75% | 90.44 | 85.56 | 4.88 | 87.04 | 84.11 | 2.92 |
| 100% | 89.3 | 86.7 | 2.6 | 92.30 | 87.70 | 4.60 |

## A.6 MORE ABLATION STUDIES

In 5, we studied the effect of different components of FairerCLIP such as $\text{Dep}(Z, Y)$, $\text{Dep}(Z_I, Z_T)$, and updating the prediction of the target labels in each iteration. Here, we add another ablation study on the effect of our control hyper-parameters, $\tau$ and $\tau_z$ on the performance of the method.

**Effect of $\tau$ and $\tau_z$:** We illustrate the performance of average accuracy and worst group's accuracy for varying values of $\tau$ and $\tau_z$ in Figure 6. First, as $\tau$ and $\tau_z$ vary, there is a smooth change in the average and worst group accuracy, which demonstrates the stability of FairerCLIP. Second, as $\tau$ increases, the worst group's accuracy also improves, which alludes to the effectiveness of $\tau$ as a control for the degree of debiasing. However, when $\tau$ reaches a certain value, a further increase in its value leads to a degradation in the worst group's accuracy. Similarly, $\tau_z$ also plays a gradual but noticeable effect on improving both the average and worst group accuracy for a given value of $\tau$.

## A.7 COMPARING FEATURES OF CLIP VIT-L/14 AND CLIP RESNET-50

To compare the features of CLIP ViT-L/14 and CLIP ResNet-50, we measure the amount of information from the target attribute and sensitive attribute contained in their generated representations. The embedded information is measured in terms of statistical dependency between the features and

Table 8: Comparison between CLIP ViT-L/14 and CLIP ResNet-50 in terms of the amount of the information from $Y$ and $S$ that each of them can embed into their feature space on the Waterbirds.

|  | $(X, Y)$ | | $(X, S)$ | |
| --- | --- | --- | --- | --- |
|  | HSIC | KCC | HSIC | KCC |
| CLIP ViT-L/14 | 0.1849 | 0.8267 | 0.2392 | 0.8661 |
| CLIP ResNet-50 | 0.1423 | 0.7556 | 0.3823 | 0.8861 |

their ground-truth $Y$ and $S$ labels. To calculate these dependencies, Hilbert-Schmidt independence criterion (HSIC) (Gretton et al., 2005) and Kernel Canonical Covariance (KCC) Bach & Jordan (2002) are used. Tab. 8 compares these two CLIP models. From the table, we can observe that CLIP ViT-L/14 embeds more information about the target $Y$ while containing less information about $S$ which indicates that the former provides better features for the Waterbirds dataset.

## A.8    COMPARING MORE THAN 100 ZERO-SHOT CLIP MODELS ON CFD

As we mentioned in Section 4.2.2, zero-shot classification on CFD is a difficult task for the OpenAI CLIP model. In Figure 7, we show that a majority of other publicly available CLIP models suffer similarly on CFD. In fact, several models achieve only near-zero WG accuracy, irrespective of their parameter count and the training dataset.

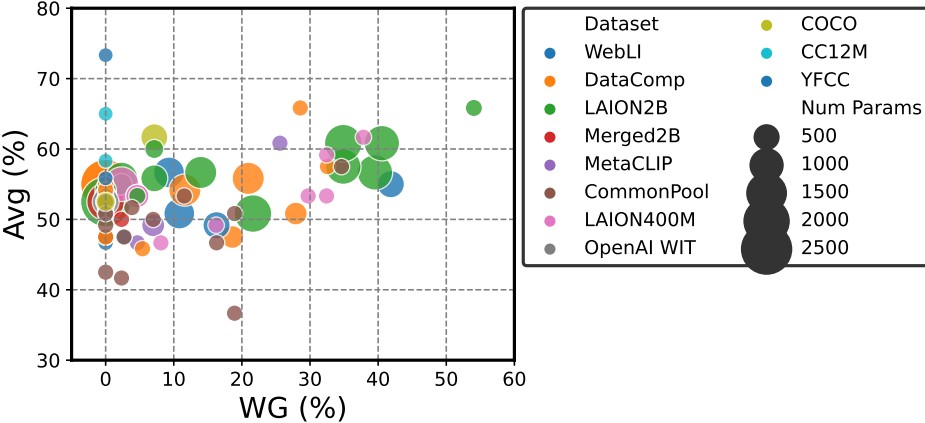

Figure 7: Comparison of more than 100 publicly available CLIP models zero-shot performance on CFD dataset. Colors show the pre-trained dataset and sizes show the number of parameters of each model.

