# OpenReview forum: "FairerCLIP: Debiasing CLIP's Zero-Shot Predictions using Functions in RKHSs"
_ICLR.cc/2024/Conference — ICLR 2024 poster_

### Official Review · Reviewer_75gn · 2023-10-25

**Soundness:** 3 good
**Presentation:** 3 good
**Contribution:** 3 good
**Rating:** 6
**Confidence:** 4

**Summary:**

In this work, the authors address the fairness problems in visual-language models (VLMs). More specifically, they propose a framework for jointly debiasing VLMs’ image and text representations. The proposed framework utilizes an alternating optimization-based approach to debias VLM representations. The authors evaluate their work using several datasets and show that FairVLM alleviates the debiasing problems of vanilla VLMs.

**Strengths:**

1. Compared to previous debiasing works in VLMs, FairVLM results in the debiasing of both image and textual representations.

2. FairVLM is agnostic to the availability of data labels and can generate debiased representations with or without labels.

3. Using the properties of RKHS in mapping the original VLM representations to a debiased space is interesting.

**Weaknesses:**

1. The authors did not compare their results with unimodal baselines, i.e., techniques that debias only the image/text representations of the VLM.

2. While the authors argue that RKHS has nice universal approximation properties, it's unclear how and why they aid in debiasing the original representations.

**Questions:**

Please refer to the weakness section for more details.

---

> ### Author Response · Authors · 2023-11-16
> **Response to comments of Reviewer 75gn**
>
> We thank the reviewer for their feedback. Below are our responses to individual questions.
>
> > ****W1**** The authors did not compare their results with unimodal baselines, i.e., techniques that debias only the image/text representations of the VLM.
> >
>
> ****A1**** We compared our method with various baselines, both with and without ground truth $Y$. Among those, Orth-Cali (Chuang et al., 2023) only debiases the VLM’s text representations, and Contrastive Adapter (Zhang & Ré, 2022) debiases only the VLM’s image representations.
>
> Moreover, FairVLM itself can be adapted to debias only one of the modalities by setting $\tau_I$ or $\tau_T$ to zero. In either case, we will have a closed-form solution for the encoder parameters in a single shot and will not need to perform any alternating optimization.
>
> Our motivation for debiasing text and image encoders stems from prior studies demonstrating bias in both image encoders [R1-R3] and text encoders [R4-R8]. Subsequently, to achieve our goal of proposing a method that can perform under different settings and conditions of debiasing, we made FairVLM controllable so that it can easily be adapted to an unimodal debiasing setting by only setting $\tau_I$ or $\tau_T$ to zero.
>
> > ****W2**** While the authors argue that RKHS has nice universal approximation properties, it's unclear how and why they aid in debiasing the original representations.
> >
>
> ****A2**** The key to the effectiveness of FairVLM is debiasing with HSIC type measure of dependence, which measures non-linear dependence. HSIC is typically estimated through RKHS. And, as we explain on page 4, under the Choice of Encoder paragraph, RKHS lends itself to closed-form solutions for the optimization problem, which is vital when optimizing competing objectives. SGD-type algorithms are not well suited in such cases, as is common in GANs and other adversarial learning methods. We repeat our description in the paper below for completeness:
>
> “Our choice of RKHS is motivated by several reasons. Debiasing is inherently an optimization problem with multiple competing objectives. In such cases, optimization is the primary bottleneck rather than model expressivity. This was also observed in Sadeghi et al. (2022). The closed-form solution afforded by our approach mitigates the optimization challenges (Sec. 4.3 and App. A.6).”
>
> Moreover, in Section 3.1 and Figure 4, we provide a geometric illustration of how FairVLM works. Using a universal RKHS like RBF kernel maps the data into infinite dimensional space where all attributes are perhaps amenable to disentangling through linear mappings, thus allowing us to estimate the correct direction of $Y$ and $S$ labels. In this space, a linear correlation can capture all types of dependencies, i.e., all linear and non-linear dependencies.
>
> If our responses addressed your initial comments, please consider raising the score.

---

> > ### Comment · Reviewer_75gn · 2023-11-18
> > **Rebuttal response**
> >
> > Thank you for your detailed rebuttal response. I will wait for the comments from all other reviewers. For now, I will keep my rating of "weak accept."

---

### Official Review · Reviewer_9VkB · 2023-10-30

**Soundness:** 3 good
**Presentation:** 3 good
**Contribution:** 3 good
**Rating:** 6
**Confidence:** 3

**Summary:**

This paper suggested a VLM debiasing method that remove bias in visual and text representations jointly by using reproducing kernel Hilbert space and deploying statistical dependency measure in RKHS. This enables to considering nonlinearity between the representation and the attribute. This paper provides a closed form solution for such formulation, and theoretical analysis on its complexity. Experiments show that the suggested method can work well both with and without true labels setting.

**Strengths:**

- Paper is well written and organization is clear.
- While previous methods mainly depend on the linearity of representation and the attribute, the suggested method can overcome such linearity assumption using RKHS.
- The suggested method is practically competitive in both of settings — w or w/o labels.
- Ablation experiments cover various scenarios, providing solid understanding of variables that affect performance.

**Weaknesses:**

While one of core features of VLM is zero-shot classification, the suggested method still requires parameter tuning (RBF kernel parameter) and stacking test data, which could be a limitation of the method.

**Questions:**

- Can this method be extended to a single point debiasing? (i.e. a single point inference for online prediction?)
- It looks like FairVLM sacrifices average scores more than other methods such as Contrastive Adapter in Table 2 w/ labels result. Furthermore, FairVLM works similarly in w/labels and w/o labels in CelebA. What’s a good interpretation on this?
- Table 3 CFD results look interesting! Does it imply FairVLM has its strength when the number of training samples is limited? How do other zero-shot methods in CFD dataset?
- Is there any convergence guarantee for Algorithm 1 (FairVLM Training Without Labels)? Also, I am wondering if there can be a failure mode that the errors in the initialization step propagate further in iteration steps.
- Possibly related works
    - Chen, A. S., Lee, Y., Setlur, A., Levine, S., & Finn, C. (2023). Project and Probe: Sample-Efficient Domain Adaptation by Interpolating Orthogonal Features. *arXiv preprint arXiv:2302.05441*.
    - Adila, D., Shin, C., Cai, L., & Sala, F. (2023). Zero-Shot Robustification of Zero-Shot Models With Foundation Models. *arXiv preprint arXiv:2309.04344*.
    - An, B., Zhu, S., Panaitescu-Liess, M. A., Mummadi, C. K., & Huang, F. (2023, July). More Context, Less Distraction: Improving Zero-Shot Inference of CLIP by Inferring and Describing Spurious Features. In *Workshop on Efficient Systems for Foundation Models@ ICML2023*.

---

> ### Author Response · Authors · 2023-11-16
> **Response to comments of Reviewer 9VkB (Part 1)**
>
> We thank the reviewer for their feedback. In response to the question regarding the performance of the other zero-shot models on the CFD experiment, we now evaluated more than 100 publicly available VLMs in addition to CLIP on CFD. The results provided more insight into understanding the challenging nature of this particular CFD scenario. In the following, we respond to each comment.
>
>
>
> > ****W1**** While one of core features of VLM is zero-shot classification, the suggested method still requires parameter tuning (RBF kernel parameter) and stacking test data, which could be a limitation of the method.
> >
>
> ****A1**** Please recall that we consider two scenarios. One where we have ground truth $Y$ labels and the other without $Y$ labels. In both cases, ground truth $S$ labels are not available.
>
> In both cases, we have a train-test split where we learn on the train split and evaluate on the test split. So, in both cases, there is no stacking of test data, and we can evaluate a single test sample at a time. We stack the train data in both cases and select the RBF kernel parameter through cross-validation. This experimental design choice was motivated by supervised baselines, which train with ground truth $Y$. We do not perform any online updates for FairVLM.
>
> The reviewer’s comment suggests an alternative scenario where one learns FairVLM at test time and applies the learned model to the same test samples. In this scenario, yes, we have to stack the test data. However, even in this scenario, it is possible to pre-train FairVLM on a separate set of training samples (only images, without labels) obtained through other means and apply it to the test sample without training. For a completely novel task where samples for training (only images, without labels) cannot be obtained, we will have to stack test data for learning FairVLM. And, as we demonstrate, FairVLM can learn from limited data (e.g., CFD dataset with 597 samples).
>
> > ******Q1******  Can this method be extended to a single point debiasing? (i.e. a single point inference for online prediction?)
> >
>
> ****A2**** Yes, as we explained in response to the previous comment, we do not train at test time but instead train on a separate dataset. So at test time, we can debias a single point.
>
> > ****Q2**** It looks like FairVLM sacrifices average scores more than other methods such as Contrastive Adapter in Table 2 w/ labels result. Furthermore, FairVLM works similarly in w/labels and w/o labels in CelebA. What’s a good interpretation on this?
> >
>
> ****A3**** We note that the Contrastive Adapter is tailored for the supervised setting and employs a complex anchor-based loss function. So, it is critically dependent on access to ground-truth labels. Therefore, it cannot be easily adapted to scenarios without ground-truth labels. In contrast, FairVLM is designed to be flexible and equally applicable to settings with and without ground truth labels without customization. The anchor-based loss function in the Contrastive Adapter can be integrated into FairVLM to improve performance under supervised learning at the cost of losing the flexibility in adapting to all different scenarios in terms of label availability.
>
> It is worth mentioning that even with this flexible approach, FairVLM can outperform Contrastive Adapter in both the CelebA and Waterbirds datasets of the CLIP ViT-L/14  model in terms of worst-group accuracy (WG) and Gap (Table 2) and also the Chicago Face Database (Table 3 (right)). Furthermore, since FairVLM can also mitigate the representation's unfairness and spurious correlations, it outperforms other baselines, including Contrastive Adapter, in fairness experiments such as CelebA for high cheekbones (Table 1).
>
> In the experiments reported in Table 2 on the CelebA dataset, the target attribute ($Y$) is blonde hair or not. Since CLIP models are trained on large-scale and diverse datasets, their zero-shot prediction for the worst group on easy-to-predict attributes like blonde hair is closer to the average across all groups. So, there is no significant performance gap between learning FairVLM with and without labels.
>
> To support this claim, we extract the zero-shot prediction accuracy of the target labels presented in Table 2 and report it below again (Table R1). We observe for CelebA that the gap between Avg. and the worst group is lower than Waterbirds, and the average accuracy is reasonably high (~87%). So FairVLM’s performance with and without labels is similar for CelebA, while the difference is larger for Waterbirds.
>
> ********Table R1:  $\hat{Y}$ zero-shot prediction accuracy by CLIP ViT-L/14********
>
> |  | Avg. | WG |
> | --- | --- | --- |
> | Waterbirds | 84.4 | 45.3 |
> | CelebA | 87.6 | 72.8 |

---

> > ### Author Response · Authors · 2023-11-16
> > **Response to comments of Reviewer 9VkB (Part 2)**
> >
> > > ******Q3****** Table 3 CFD results look interesting! Does it imply FairVLM has its strength when the number of training samples is limited? How do other zero-shot methods in CFD dataset?
> > >
> >
> > ****A4**** Our motivation for conducting this experiment was to evaluate FairVLM and other baselines in an extremely challenging setting. As mentioned in the paper, the CFD experiment has two characteristics that make it difficult.
> >
> > 1. The number of training samples is minimal (597 samples), which might not be sufficient for models with a large number of parameters.
> > 2. The performance of the zero-shot classifier for this task suggests that the image and text features from CLIP are not aligned with each other for the worst group, leading to incorrect predictions for $\hat{S}$ and $\hat{Y}$. In response to the second question, we now evaluated the zero-shot prediction performance of 108 publicly available models from the OpenCLIP repository to verify if CLIP is an outlier. Below, we present the results of a select few models in Table R1. The results clearly indicate that this task is challenging for most of the VLMs, as reflected in their low average and worst group accuracy. The results for the entire list of 108 models are too long to include in the paper. So, we will make them available along with the code in a CSV file.
> >
> > ****************Table R3: CFD results for other VLMs****************
> >
> > | OpenCLIP Models | #Parameters (Millions) | Pretrained Dataset | WG | Avg | Gap |
> > | --- | --- | --- | --- | --- | --- |
> > | ViT-SO400M-14-SigLIP-384 | 877.96 | WebLI | 10.81 | 50.83 | 40.02 |
> > | ViT-bigG-14-CLIPA-336 | 2,517.76 | DataComp1B | 0 | 55.0 | 55.0 |
> > | ViT-SO400M-14-SigLIP | 877.36 | WebLI | 0 | 55.0 | 55.0 |
> > | ViT-H-14-CLIPA-336 | 968.64 | DataComp1B | 11.0 | 54.17 | 43.17 |
> > | convnext_xxlarge | 1,200.58 | LAION2B | 34.88 | 57.5 | 17.14 |
> > | EVA01-g-14 | 1,136.44 | LAION400M | 2.33 | 55.0 | 52.67 |
> > | RN101 | 119.69 | OpenAI | 0 | 52.5 | 52.5 |
> > | RN50 | 102.01 | OpenAI | 0 | 50.83 | 50.83 |
> >
> > > ****Q4**** Is there any convergence guarantee for Algorithm 1 (FairVLM Training Without Labels)? Also, I am wondering if there can be a failure mode that the errors in the initialization step propagate further in iteration steps.
> > >
> >
> > ****A5**** There is no theoretical convergence guarantee for Algorithm 1 apart from the closed-form solvers in each iteration. But empirically, we observed that the algorithm converges to local optima across all our experiments. For tasks where $Y$ labels are available, FairVLM typically converged in just one iteration and saturated its best performance. In the case w/o ground truth $Y$ labels, we found that it converged in less than five iterations when its best performance was also observed. In this case, the extra iterations might be since we also update the target attribute’s pseudo-labels in each iteration.
> >
> > Across all the experiments presented in the paper, we did not notice the algorithm fail, at least not catastrophically, as the other approaches did in the CFD experiment. Even in the CFD dataset case, where the error in the initial zero-shot predictions is very high, we did not observe FairVLM fail as the results demonstrate. However, our observations do not preclude possible failures, either due to the scenario described by the reviewer or otherwise. We did not encounter any failures in *our* experiments.
> >
> > > ******Q5****** Possibly related works …
> > >
> >
> > ****A6**** Thank you for the suggestions. These papers seek to mitigate only spurious correlations and focus on domain generalization or robustness. (1) uses partial labels while we consider full or no labels. (2) does not use labels for debiasing, but performance is lower than some of the baselines we already compare against. (3) is a training-free debiasing approach. We have added them to the updated paper at appropriate places.
> >
> > If our responses addressed your initial comments, please consider raising the score.

---

> ### Comment · Reviewer_9VkB · 2023-11-18
>
> Thank you for the detailed clarification and updated results; most of my initial comments are nicely addressed. I have a follow-up comment on A2.
>
> ```
> A2. Yes, as we explained in response to the previous comment, we do not train at test time but instead train on a separate dataset. So at test time, we can debias a single point.
> ```
>
> My comment was about when such a separate dataset is unavailable, and we want to do zero-shot prediction for a single test data point. In my understanding, the answer is no. However, acquiring unlabeled dataset is typically not challenging, the proposed method remains valuable.
>
> While I find the technical contribution of this paper to be sound and solid, I suggest the authors provide further clarification regarding the specific category and limitations of their proposed method. When true labels are available, the method can be seen as an adapter approach that uses kernel methods outlined in the paper. In this view, it might be inappropriate to label it as "zero-shot" prediction, since the method requires a sort of adapter training on the specific task. In the absence of true labels, the method can be viewed as a test-time adaptation using pseudo-labels. One of potential limitation is that applying it to a single or a few test data points can be challenging.
>
> I have increased my score to 6.

---

> > ### Author Response · Authors · 2023-11-20
> > **Response to the official comment of Reviewer 9VkB**
> >
> > **Thank you for your response.**
> >
> >
> > > ******Q1****** My comment was about when such a separate dataset is unavailable, and we want to do zero-shot prediction for a single test data point. In my understanding, the answer is no. However, acquiring unlabeled dataset is typically not challenging, the proposed method remains valuable.
> > >
> >
> > ****A7**** Learning to debias and make a prediction from a single sample is extremely challenging, not only for FairVLM but for any other approach that has to learn from one sample without labels. This can be resolved through learning complementary concepts like test-time training with data augmentation. From a single sample, multiple augmentations of the sample can be obtained, and FairVLM could be trained on this and then utilized for making a prediction.
> >
> > So, in principle, we argue that learning to debias and making a prediction from a single test data point is *feasible. S*till, the design of such a solution and its effectiveness must be thoroughly examined through careful evaluation. We believe this warrants a new and independent study and is beyond the scope of this paper. We would like to sincerely thank the reviewer for this line of inquiry. It provided us with some food for thought and opened up a possible new research direction.
> >
> > Meanwhile, we would like to bring the reviewer’s attention to Table 7 in section A.5 (page 17), where we studied the effect of training from a limited amount of data. As the results suggest, even with 5% data (~200 samples on Waterbird), FairVLM (FairerCLIP) has a considerably higher average and worse group accuracy than the original CLIP model. So it is plausible that FairVLM trained with a few test samples with data augmentation can make the original CLIP predictions more fair.
> >
> > > ******Q2****** While I find the technical contribution of this paper to be sound and solid, I suggest the authors provide further clarification regarding the specific category and limitations of their proposed method. When true labels are available, the method can be seen as an adapter approach that uses kernel methods outlined in the paper. In this view, it might be inappropriate to label it as "zero-shot" prediction, since the method requires a sort of adapter training on the specific task. In the absence of true labels, the method can be viewed as a test-time adaptation using pseudo-labels. One of potential limitation is that applying it to a single or a few test data points can be challenging.
> > >
> >
> > ****A8**** We agree with the reviewer that using the phrase "zero-shot prediction" for the combined frozen CLIP + FairVLM approach is, perhaps, not very precise, at least in the scenario where training labels are available. For further clarification, the scenario with labels could be viewed as a semi-supervised scenario since $S$ labels are not available and estimated through zero-shot prediction from the CLIP model. To reflect this, we made the following changes: modified the title, and replaced "zero-shot prediction" with "prediction" in parts of the paper where appropriate to avoid having to introduce two different terms for predictions made by learning with and without labels. All the changes are colored blue in the revised version of the paper.
> >
> > As we elaborate in A7, applying the method to a single or a few test samples is, in principle, *feasible,* but is not the focus of this paper. We added a footnote at the bottom of page 3 to reflect the same.
> >
> > > ******Q3****** I have increased my score to 6.
> > >
> >
> > **A9** We want to thank you for revising your initial score.
> >
> > We sincerely appreciate the time and effort you dedicated to reviewing our paper and providing insightful suggestions. Your valuable input has improved the clarity of the paper.
> >
> > Please let us know if our responses and edits to the paper did not sufficiently address your query about "clarification regarding the specific category and limitations" of the paper.

---

### Official Review · Reviewer_2eeh · 2023-10-31

**Soundness:** 4 excellent
**Presentation:** 3 good
**Contribution:** 4 excellent
**Rating:** 8
**Confidence:** 1

**Summary:**

This paper introduces FairVLM, a novel approach designed to address bias in zero-shot predictions made by VLMs. FairVLM demonstrates versatility in mitigating bias arising from two primary sources: spurious correlations and intrinsic dependencies within the data. Moreover, it offers the flexibility to be trained with or without the presence of ground-truth labels.

**Strengths:**

1. The paper demonstrates that a single general method can debias the image and text features of VLMs under different scenarios more effectively than specialized solutions for each scenario. The scenarios include accounting for both spurious correlations and intrinsic dependencies, learning with and without ground-truth labels, and learning from small and medium-sized datasets
2. The words are fluent.

**Weaknesses:**

1. The experiment results on the datasets (w/ labels) are not good enough.
2. The pare of the method is too complex.

**Questions:**

I don't have any questions because I cannot understant the method part.

**Details Of Ethics Concerns:**

no ethics concerns here

---

> ### Author Response · Authors · 2023-11-16
> **Response to comments of Reviewer 2eeh**
>
> We thank the reviewer for their feedback. Below are our responses to individual questions.
>
> > ******W1****** The experiment results on the datasets (w/ labels) are not good enough.
> >
>
> ****A1**** As pointed out by the reviewer as a strength, one of the main contributions of FairVLM is its ability to adapt to different settings to mitigate various types of biases. In this paper, we have conducted several experiments on FairVLM, including mitigating spurious correlation, mitigating intrinsic dependencies, and conducting experiments on datasets with limited training samples and both w/ and w/o ground-truth labels. In all of these settings, FairVLM either outperforms other baselines, e.g.,
>
> - In Table 2, WG and Avg. increased for all experiments (except CelebA on CLIP ResNet-50) in w/o label settings,
> - Chicago Face Dataset (CFD) in Table 3 (right)
> - Reduce MaxSkew fairness metric for the FairFace dataset in Table 3 (left).
> - Reduce the fairness metric EOD on the CelebA dataset in Table 1.
> - Improve the WG accuracy and decrease the Gap for experiments on CLIP ViT-L/14 in Table 2.
>
> Or it is second best in other experiments except for CelebA for CLIP ResNet-50 in w/ ground-truth labels.
>
> Furthermore, FairVLM enjoys computational advantages for training (see Table 4). On the Waterbirds dataset, FairVLM is 40 times more efficient than Contrastive Adapter, which has some accuracy advantages over FairVLM in two of the experiments on CLIP ResNet-50 and is five times more efficient than the other baselines. On the CelebA dataset that contains more samples, FairVLM is more than ten times faster than the model with the best accuracy, DRF (Upsample), and about 100 times faster than the Contrastive Adapter, which is the second-best method in terms of accuracy in that specific setting. It is worth mentioning that since the FairVLM employs kernel methods in its encoders, it can be scaled to medium-sized datasets using Random Fourier Features [R1], while the other baselines are not efficient even for the relatively small-sized datasets (CelebA and Waterbirds), we evaluated.
>
> In conclusion, based on the information provided above, we maintain a robust belief that **suboptimal outcomes against specialized solutions occur in only two out of thirteen experiments.** These results amply demonstrate the overall value and effectiveness of FairVLM's contributions.
>
> > ******W2****** The pare of the method is too complex.
> >
>
> ****A2**** Our solution employs kernel methods, which are part of the standard machine learning toolbox. Fig 4 in Section 3.1 provides a geometric illustration of FairVLM. Moreover, Fig. 2 provides an overview of the training and inference phases of the proposed method, and Fig. 3 explains our training algorithm as a figure.
>
>
> If our responses addressed your initial comments, please consider raising the score.
>
>
> ## ********************References********************
>
> [R1] Ali Rahimi and Benjamin Recht. Random features for large-scale kernel machines. Advances in neural information processing systems, 20, 2007.

---

### Official Review · Reviewer_p1sH · 2023-11-01

**Soundness:** 3 good
**Presentation:** 3 good
**Contribution:** 3 good
**Rating:** 6
**Confidence:** 4

**Summary:**

This paper proposes FairVLM, which is an additional module top on the frozen CLIP features, to de-bias the prediction. Using the frozen CLIP vision and text encoders, the proposed method extracts visual and texture features, and lets them be de-biased using the Hilbert-Schmidt Independence Criterion (HSIC), a famous approach in de-bias literature. Instead of using the original HSIC, this paper proposes to use a simplified version following Sadeghi et al. The proposed simplified HSIC provides a closed-form solution to the additional feature encoders when the features are fixed. To make the method efficient, this paper proposes to approximate Cholesky decomposition using random Fourier features (RFF), resulting in reducing the computational complexity from $O(n^3)$ to $O(n^2)$. Experimental results show that the proposed method shows the effectiveness of the proposed method in both intrinsic dependency (i.e., fairness scenario) and spurious correlation.

**Strengths:**

HSIC is a promising approach to achieving de-biased representations, as many previous studies have observed. This paper successfully brings the advantage of HSIC to CLIP feature refinement tasks. I also think that this paper has a non-trivial contribution to introducing the closed-form solutions used for updating the parameters, including the approximated version of Cholesky decomposition using RFF. Straightforwardly, an iterative algorithm using a closed-form solution will converge much faster than gradient-based algorithms, as shown in classic machine learning studies, such as ADMM [R1]

- [R1] Boyd, Stephen, et al. "Distributed optimization and statistical learning via the alternating direction method of multipliers." Foundations and Trends® in Machine learning 3.1 (2011): 1-122.

Combining two good properties (HSIC, a promising approach, and an efficient update algorithm using a closed-form solution), the proposed method shows promising performances on the given evaluation benchmarks.

**Weaknesses:**

### Scope of the paper

The terminology "VLM" is misused in this paper. VLM literally includes a vast area of models trained with vision and language. For example, visual-question answering (VQA) is a VLM model, vision-language pre-training (VLP) with cross-attention transformers (such as ViLBERT [R2], ViLT [R3], Align [R4], VinVL [R5], ALBEF [R6], BLIP [R7]) is VLM, multi-modal generation models, such as dall-e 1, 2 and 3, stable diffusion or dreambooth, are VLM, and recent language-model combined vision models, such as BLIP2 [R8], Fromage [R9], GPT-4, are VLM. (I omitted some famous works, such as dall-e, SD, GPT ...).

- [R2] Lu, Jiasen, et al. "Vilbert: Pretraining task-agnostic visiolinguistic representations for vision-and-language tasks." Advances in neural information processing systems 32 (2019).
- [R3] Kim, Wonjae, Bokyung Son, and Ildoo Kim. "Vilt: Vision-and-language transformer without convolution or region supervision." International Conference on Machine Learning. PMLR, 2021.
- [R4] Jia, Chao, et al. "Scaling up visual and vision-language representation learning with noisy text supervision." International conference on machine learning. PMLR, 2021.
- [R5] Zhang, Pengchuan, et al. "Vinvl: Revisiting visual representations in vision-language models." Proceedings of the IEEE/CVF conference on computer vision and pattern recognition. 2021.
- [R6] Li, Junnan, et al. "Align before fuse: Vision and language representation learning with momentum distillation." Advances in neural information processing systems 34 (2021): 9694-9705.
- [R7] Li, Junnan, et al. "Blip: Bootstrapping language-image pre-training for unified vision-language understanding and generation." International Conference on Machine Learning. PMLR, 2022.
- [R8] Li, Junnan, et al. "Blip-2: Bootstrapping language-image pre-training with frozen image encoders and large language models." arXiv preprint arXiv:2301.12597 (2023).
- [R9] Koh, Jing Yu, Ruslan Salakhutdinov, and Daniel Fried. "Grounding language models to images for multimodal generation." arXiv preprint arXiv:2301.13823 (2023).

However, the scope of this paper is very narrow compared to the entire VLM family. I feel that the title "FairVLM" and the terminology "VLM" are too much overclaimed, and a reader can misunderstand the focus of this paper. I think this paper should tone down its contribution and focus more specifically. This paper only targets a feature refinement method top on the frozen CLIP model, and the comparison methods are also methods using frozen CLIP encoders. Note that it is still a narrow topic in "de-biasing CLIP zero-shot prediction" because a number of works focus on the fine-tuning strategy [R10-12]. On the other hand, this paper relies on the pre-trained feature encoders that may weakens its contribution.

- [R10] Wortsman, Mitchell, et al. "Robust fine-tuning of zero-shot models." Proceedings of the IEEE/CVF Conference on Computer Vision and Pattern Recognition. 2022.
- [R11] So, Junhyuk, et al. "Geodesic multi-modal mixup for robust fine-tuning." arXiv preprint arXiv:2203.03897 (2022).
- [R12] Vogt-Lowell, Kevin, et al. "Robust Fine-Tuning of Vision-Language Models for Domain Generalization." IEEE High Performance Extreme Computing Conference (HPEC). 2023.

### Missing related works

Including [R1-12] in the first comment, there are a number of missing related works that should be discussed. For example, HSIC is a popular approach in de-biasing studies [R13, R14]. R13 directly optimizes the HSIC between features and sensitive attribute labels, similar to Dep(Z, Y) in the paper; R14 optimizes the HSIC between biased features and target features to avoid using sensitive attribute labels Y. If we extend our viewpoint to RHKS, there is work using MMD to achieve fairness [R15]. I omitted many HSIC-based regularization methods that could be related to this work, but if possible, it would be great to add more citations for methods using HSIC.

- [R13] Quadrianto, Novi, Viktoriia Sharmanska, and Oliver Thomas. "Discovering fair representations in the data domain." Proceedings of the IEEE/CVF conference on computer vision and pattern recognition. 2019.
- [R14] Bahng, Hyojin, et al. "Learning de-biased representations with biased representations." International Conference on Machine Learning. PMLR, 2020.
- [R15] Jung, Sangwon, et al. "Fair feature distillation for visual recognition." Proceedings of the IEEE/CVF conference on computer vision and pattern recognition. 2021.

Also, in terms of employing pseudo-labels for the de-biasing optimization, I think this paper is also related to [R16]; where R16 is based on semi-supervised learning without the pseudo-label refinement process.

- [R16] Jung, Sangwon, Sanghyuk Chun, and Taesup Moon. "Learning fair classifiers with partially annotated group labels." Proceedings of the IEEE/CVF Conference on Computer Vision and Pattern Recognition. 2022.

### Unclear or missing details

While reading the paper, I had trouble understanding the details of the paper. For example, it is still unclear to me how the "encoder" works. I presume that the encoder parameter $\Theta$ is a linear projection and an RBF kernel is used for computing HSIC, but it is unclear. Also, there is no description of $r$ (r becomes the dimensionality of the generated representation) and the value of $r$ as well. It means that this paper does not provide any detail of RKHS hyperparameters, such as the dimensionality of the projection layer and the hyperparameter of the kernel method. Similarly, I cannot find any detail of the choice of the learning hyperparameters, such as the number of iterations, and batch size. It means that it is impossible or extremely difficult to reproduce the results in this paper. Overall, this paper is very hard to understand the method details and implementation details, although I think this paper has certain contributions in terms of the methodology development.

**Questions:**

I think the technical contribution of this paper is sound and empirical evaluation results look reasonable. However, this paper has a critical problem in its writing and presentation, including many missing related works and details. Please check my initial review and respond to my concerns. Specifically, I would like to clarify all the details of how the method is implemented and trained in the revised version, which is not presented in the initial version. I think the initial version is improper to be published as an ICLR paper, but my concern is mostly around the writing, that could be improved during the revision period. I presume that the revised manuscript will need significant efforts, but mainly in the presentation, rather than technical enhancement. Hence, if the revised manuscript is sound and can resolve my initial concerns, I am willing to update my score.

---

> ### Author Response · Authors · 2023-11-16
> **Response to comments of Reviewer p1sH (Part 1)**
>
> We thank the reviewer for their feedback. Below are our responses to individual questions.
>
> > ****W1****  The terminology "VLM" is misused in this paper. VLM literally includes a vast area of models trained with vision and language. I feel that the title "FairVLM" and the terminology "VLM" are too much overclaimed, and a reader can misunderstand the focus of this paper.
> >
>
> ****A1**** We thank the reviewer for detailing the generality of the term “VLM.” We narrowed the focus of our presentation to “debiasing CLIP embeddings for zero-shot predictions.” We renamed our approach to FairerCLIP and rephrased all instances of VLM with CLIP or zero-shot predictions with CLIP, as appropriate. We also changed the paper's title to “FairerCLIP: Mitigating Bias in CLIP Representations for Zero-Shot Prediction.”
>
> **To avoid confusing the other reviewers, we continue to use the term “FairVLM” in the rebuttal.**
>
> > ****W2**** I think this paper should tone down its contribution and focus more specifically. This paper only targets a feature refinement method top on the frozen CLIP model, and the comparison methods are also methods using frozen CLIP encoders.
> >
>
> ****A2**** We have renamed our proposed method changed the title, and toned down our contributions and focus to “debiasing CLIP embeddings for zero-shot predictions.” We hope that our changes address the reviewer’s concerns.
>
> While our approach operates on a frozen CLIP model, some of the baselines we compare against perform fine-tuning on at least one layer in CLIP if not all.
>
> - **ERM Linear Probe** fine-tunes the last linear layer of CLIP.
> - **ERM Adapter** adds new linear layers and fine-tunes them
> - **DFR** fine-tunes the last layer of the CLIP
>
> Despite not fine-tuning CLIP, FairVLM outperforms those that fine-tune CLIP, reiterating its effectiveness.
>
> > ******W3****** Note that it is still a narrow topic in "de-biasing CLIP zero-shot prediction" because a number of works focus on the fine-tuning strategy [R10-12]. On the other hand, this paper relies on the pre-trained feature encoders that may weakens its contribution.
> >
>
> ****A3****  Respectfully, we disagree with the reviewer that not fine-tuning the CLIP backbones weakens our contributions since. On the contrary, as we discuss below, operating directly on the features without fine-tuning is a more practical solution, especially under limited resources.
>
> First, fine-tuning requires heavy computation and typically more data for fine-tuning compared to not resorting to fine-tuning. These may not be available in many practical scenarios, especially zero-shot prediction scenarios. Without sufficient care, fine-tuning may lead to severe over-fitting. Second, we already compare to baselines (ERM Linear Probe, DFR) that perform fine-tuning and demonstrate that FairVLM outperforms them significantly. Thirdly, FairVLM can be employed to debias feature representations in downstream tasks without access to the original images or model weights. Finally, employing fine-tuning along with FairVLM is undoubtedly feasible if required.
>
> Based on the above, debasing frozen CLIP representations is not a narrow topic but has a wide range of practical application scenarios.
>
> > ******W4****** there are a number of missing related works that should be discussed. For example, HSIC is a popular approach in de-biasing studies [R13, R14]. R13 directly optimizes the HSIC between features and sensitive attribute labels, similar to Dep(Z, Y) in the paper; R14 optimizes the HSIC between biased features and target features to avoid using sensitive attribute labels Y. If we extend our viewpoint to RHKS, there is work using MMD to achieve fairness [R15]. I omitted many HSIC-based regularization methods that could be related to this work, but if possible, it would be great to add more citations for methods using HSIC.
> >
>
> ****A4**** Thank you for providing such a comprehensive list of suggestions for related work; we really appreciate it. We added the three suggested papers (R13-R15) to the related discussion on page 3 of the paper in the **Choice of Dependence Measure** paragraph. There are potentially other methods that use HSIC as a regularizer. However, due to page limits for the main paper, we could not add more of them at this time.
>
> > ****W5**** in terms of employing pseudo-labels for the de-biasing optimization, I think this paper is also related to [R16]; where R16 is based on semi-supervised learning without the pseudo-label refinement process.
> >
>
> ****A5**** From our understanding of R16, it has access to partially annotated sensitive attributes and estimates pseudo labels for the sensitive attribute for the rest of the training samples. In contrast, in our paper, the sensitive attributes are not available for FairVLM and other baselines, which makes our setting more challenging. We have added a citation to this paper on page 5 (colored in blue).

---

> > ### Author Response · Authors · 2023-11-16
> > **Response to comments of Reviewer p1sH (Part 2)**
> >
> > > ****W6**** While reading the paper, I had trouble understanding the details of the paper. For example, it is still unclear to me how the "encoder" works. I presume that the encoder parameter $\Theta$ is a linear projection and an RBF kernel is used for computing HSIC, but it is unclear.
> > >
> >
> > **A6** For clarity, we report the details below and point to the parts of the paper where they are described.
> >
> > $Z := [Z_1, Z_2, \dots, Z_r] \text{ where } Z_j = f_j(X), f_j \in \mathcal{H}_X \text{ } \forall j=\{1,2,\dots,r\}$ (this is in Equation 5): $Z$ is the generated representation
> >
> > $\mathbf{f}(X) = \mathbf{\Theta}[k_x(x_1, X), \dots, k_x(x_n, X)]^T, \text{ where } \mathbf{\Theta} \in \mathbb{R}^{r\times n}$, $k_x(\cdot,\cdot)$ is the kernel and $n$ is the number of training samples. (this is in Theorem 2): Here $\textbf{f}$ is the encoder which is a linear projection on the RBF kernel.
> >
> > $\text{Dep}(Z, S) :=\sum_{j=1}^r \sum_{\beta \in \mathcal U_S } \mathbb{C}\text{ov}^2\left(Z_j, \beta(S)\right)$ (this is in Equation 1): This is the definition of the simplified version of HSIC.
> >
> > $\text{Dep}(Z, S) := \frac{1}{n^2}\left\|   \mathbf{\Theta}  \mathbf{K}_X  \mathbf{H}  \mathbf{L}_S \right\|^2_F$ (this is in Equation 2): If we plug $Z$ from the second equation into the third equation we will get this compact formula for calculating Dep between the representation and the sensitive attribute.
> >
> > Here $Z \in \mathbb{R}^r$ is the debiased representation, and $ \mathbf{\Theta}$ are the parameters to learn.
> >
> > > ******W7****** Also, there is no description of $r$ ($r$ becomes the dimensionality of the generated representation) and the value of $r$ as well. It means that this paper does not provide any detail of RKHS hyperparameters, such as the dimensionality of the projection layer and the hyperparameter of the kernel method. Similarly, I cannot find any detail of the choice of the learning hyperparameters, such as the number of iterations, and batch size. Specifically, I would like to clarify all the details of how the method is implemented and trained in the revised version, which is not presented in the initial version.
> > >
> >
> > **A7** Thank you for noting the absence of details on choosing $r$. In all experiments, we choose $r=C-1$ where $C$ is the number of classes for label $Y$. We added this information to Appendix A.4 (Implementation Details) of the supplementary material. All other details of implementation, such as values of RFF dimensionality and control parameters, $\tau$ and $\tau_z$, were already reported in Appendix A.4 of the supplementary material.
> >
> > To train FairVLM, we compute the closed-form solution using all the training samples at once. So, there is no minibatch size as such. Our batch comprises all the training samples.
> >
> > > ******Q1****** I think the technical contribution of this paper is sound and empirical evaluation results look reasonable. However, this paper has a critical problem in its writing and presentation, including many missing related works and details.
> > >
> >
> > ******A8****** The implementation details were present in Appendix A.4. We have updated it with more information and added a table (Table 6) for clarity. Upon acceptance of the paper, we will also release our source code for reproducing all experimental results presented in the paper. We have updated the manuscript to address the reviewer’s concerns by adding more related works and narrowing the claims of the paper. We also addressed the reviewer's comments on fine-tuning methods. Please let us know if there are any unresolved comments or further concerns.
> >
> > > ******Q2****** Specifically, I would like to clarify all the details of how the method is implemented and trained in the revised version, which is not presented in the initial version.
> > >
> >
> > ******************************A9****************************** The training details and algorithm have been presented in Appendix A.1, and the implementation details have been presented in Appendix A.4. We have now updated Appendix A.4 with more information. We also added a table with implementation details in Appendix A.4 for clarity. Upon acceptance of the paper, we will also release our source code for reproducing all experimental results presented in the paper.
> >
> > > ******Q3****** I presume that the revised manuscript will need significant efforts, but mainly in the presentation, rather than technical enhancement. Hence, if the revised manuscript is sound and can resolve my initial concerns, I am willing to update my score.
> > >
> >
> > ******A10****** We have updated the manuscript to address the reviewer’s concerns. Please let us know if there are any unresolved comments or further concerns.
> >
> > If our responses addressed your initial comments, please consider raising the score.

---

> > > ### Comment · Reviewer_p1sH · 2023-11-19
> > >
> > > Thanks for your clarification and revision.
> > >
> > > - [A1, A2] I think the revised version of the paper scope makes sense to me.
> > > - [A3] I still think that remaining the CLIP backbone as frozen (including fine-tuning a small part of the model) is still somewhat in a narrow scope. However, it does not mean that this weakness is for the reject decision. As my previous comment, I think this paper has a sound technical contribution and the empirical evaluation results look reasonable.
> > > - [A4, A5] Thanks for the answers and the corresponding revision.
> > > - [A6-A9] Thanks for the clarification. I totally understand that the page limitation is strict to include all the details. I would like to suggest citing Appendix A.1, A.3 (as well as the source code repository URL) in the final main paper for clarity. It is not a mandatory request, but a mild suggestion from the reviewer.
> > > - [A10] Thanks for your hard work during the rebuttal period. I have revised my rating from 5 to 6 (considering A3), and "Presentation" from 1 to 3. Thanks again for all your efforts and clarification.

---

> > > > ### Author Response · Authors · 2023-11-20
> > > > **Response to the official comment of Reviewer p1sH**
> > > >
> > > > **Thank you for your response.**
> > > >
> > > > > [A3] I still think that remaining the CLIP backbone as frozen (including fine-tuning a small part of the model) is still somewhat in a narrow scope. However, it does not mean that this weakness is for the reject decision. As my previous comment, I think this paper has a sound technical contribution and the empirical evaluation results look reasonable.
> > > > >
> > > >
> > > > ****A11**** We wish to seek clarity on what, in the reviewer's opinion, represents the “full scope” of debiasing CLIP models to understand better why “CLIP backbone as frozen (including fine-tuning a small part of the model) is still somewhat in a narrow scope.”
> > > >
> > > > Suppose the full scope is to train a debiased CLIP from scratch or fine-tune all layers on appropriate data for every label $S$ we which to be debiased w.r.t. In that case, we respectfully disagree that it is not practically feasible, given the resources needed to train these models. And, training from scratch to debias w.r.t. all possible bias labels is also infeasible since we would always have to contend with unknown biases.
> > > >
> > > > As such, we argue that, from an application perspective, FairVLM significantly widens the scope of scenarios where debiasing methods can be employed.
> > > >
> > > > - **Bias Adaptive:** It can be customized/personalized for each task $Y$ and bias label $S$ combination. We demonstrate this through experiments on multiple datasets, tasks, and bias labels.
> > > > - **Computational Efficiency:** Can be trained efficiently without needing large-scale computational resources.
> > > > - **Number of Samples:** Table 3 (right) shows the experiment's results on the CFD dataset with a limited number of samples (597). Observe that the ERM Adapter, a fine-tuning method, worsens the accuracy of the underlying CLIP model due to overfitting. Thus, training end-to-end or training from scratch from limited samples could lead to severe overfitting and worsen both the accuracy and the bias in the model. Furthermore, as Table 7 (Section A.3) demonstrates, FairVLM exhibits good performance even when trained with only 5% data.
> > > > - **Agnostic to CLIP Model:** Since our approach operates directly on image and text features, it is agnostic to the backbone CLIP architecture. Solutions that fine-tune all or part of the model need access to the model architecture and weights and cannot be employed when the pre-trained model (architecture+weights) is unavailable.
> > > >
> > > > Based on the above, we firmly believe that the scope of practical scenarios where FairVLM can be employed is broader than methods that fine-tune a few or all layers.
> > > >
> > > > We would like to get the reviewer’s opinion on what constitutes the “full scope” of debiasing CLIP models. It would help us understand the reviewer’s perspective better and identify the limitations of our work.
> > > >
> > > > > [A6-A9] Thanks for the clarification. I totally understand that the page limitation is strict to include all the details. I would like to suggest citing Appendix A.1, A.3 (as well as the source code repository
> > > > URL) in the final main paper for clarity. It is not a mandatory request, but a mild suggestion from the reviewer.
> > > > >
> > > >
> > > > ****A12**** Thank you for the suggestion. Appendix A.1 was already referenced in the last line of Section 3, but Appendix A.3 was not directly referenced; therefore, we added a line on page 6 and referred to that section to improve clarity. Moreover, as promised in ****A9,**** we will publicly release all source code upon acceptance of the paper. We will add the link to the paper as well.
> > > >
> > > > > [A10] Thanks for your hard work during the rebuttal period. I have revised my rating from 5 to 6 (considering A3), and "Presentation" from 1 to 3. Thanks again for all your efforts and clarification.
> > > > >
> > > >
> > > > ****A13**** We thank the reviewer for their time and precise feedback. It helped improve the paper. And, based on feedback from 9VkB, we modified the title. Please let us know if you have concerns about the new title.
> > > >
> > > > If your concern about "the narrow scope of using frozen CLIP" is not sufficiently addressed or you have any other concerns, please let us know so we can address them.

---

> > > > > ### Comment · Reviewer_p1sH · 2023-11-20
> > > > >
> > > > > I already revised my score to **6** (weak accept). I don't think the paper's contribution is as strong as **8**. I still think the scope of this paper is narrower than more general methods that cover fine-tuning and from-scratch training. *It does not mean* that the contribution of this paper is not strong enough to appear in the ICLR venue, but it means that it is a "weakness" of this work. Please note that the role of reviewers is to provide enough information to chairs, including the advantages and the negatives of the paper.
> > > > >
> > > > > There could be some applications of this method, but still, this method is impossible to be applied to a from-scratching training scenario. On the other hand, previous de-biasing methods mainly tackle a from-scratching training scenario without a pre-trained model and bias labels [R13, 14 and more related works _(I will not cite them all because I don't think that this paper needs to cite the de-biasing methods)_]. Similarly, there are other CLIP variants that tackle the inherent bias of CLIP, usually based on fine-tuning [R10, R11, R12 _(Similarly, there could be more papers that I don't know, but I think this paper does not have to cite them all)_]. Resource limitation is, of course, a painful constraint to all of us (except a few large companies), but it cannot remove the weakness of this method. I still think that it is a weakness of this method, but not a significant weakness as a reject case.

---

> > > > > > ### Author Response · Authors · 2023-11-20
> > > > > > **Thank you**
> > > > > >
> > > > > > Thank you sincerely for taking the time to share your response with us.

---

### Author Response · Authors · 2023-11-16
**Summary of the Responses**

We thank all the reviewers for their valuable feedback. We incorporated the feedback in the paper and updated it with our changes highlighted in blue. We first summarize the primary concerns of the reviewers and how we addressed them:

- In response to reviewer ********p1sH’s******** concern about using the term VLM and its generality, we renamed our approach to ********************FairerCLIP.******************** We updated the paper title to “FairerCLIP: Mitigating Bias in CLIP Representations for Zero-Shot Prediction” to better align the title with our contributions. **However, to avoid confusing the other reviewers, we continue to use the term “FairVLM” in the rebuttal.**
- In response to reviewer ********p1sH’s******** concern about missing related work and implementation details, we incorporated the suggested related work into the paper. We updated Section A.3 (Implementation Details) with all implementation details. We also added Table 6 in the appendix to summarize the details for different datasets. Upon acceptance of the paper, we will release our source code for reproducing all experimental results presented in the paper.
- Reviewer ********2eeh’s******** concerns were on the performance of FairVLM in one of the experiments and the complexity of our approach. To address them, we provided a detailed comparison between FairVLM and other baselines in all our experiments. We referred to specific sections and figures in the paper to aid with an intuitive understanding of our approach without engaging the technical details.
- Reviewer ********9VkB******** mentioned the need for test-time stacking as a limitation of FairVLM. There was a potential misunderstanding. We provided a detailed explanation of how we train FairVLM offline, thus avoiding the need for test-time stacking.
- Reviewer ********9VkB******** also enquired about the experiment results on CFD. To answer this question, **we evaluated more than 100 publicly available CLIP-like models for zero-shot predictions on CFD.** We reported the results of a select few popular models in the rebuttal. Surprisingly, almost all the models had abysmal worst group accuracy on CFD. At the same time, FairVLM significantly improved the performance of representations from CLIP ViT-L/14.
- Finally, Reviewer ********75gn******** suggested comparing our results with unimodal approaches. To address it, we pointed out the baselines that debias only one of the image/text representations and compared FairVLM with them.

If there are any remaining concerns or questions that you'd like to address, please feel free to share them with us.

---

### Meta-Review · Area_Chair_Gswt · 2023-12-06

**Metareview:**

This paper seeks to debias zero-shot predictions for visual-language models like CLIP. This area has gotten popular lately due to its obvious importance. The authors propose a fairly flexible method that handles supervised and unsupervised cases and is efficient.

The main strength here is the fact that the method is compatible with labels and pseudolabels, doesn't require fine-tuning or using adapters. The downside is that the authors still require training (ie, their kernel method) which might block real-time usage, but this is not critical.

The reviewers asked for substantial improvements to clarity, which the authors have performed. The current paper is fairly clear and has a solid contribution.

**Justification For Why Not Higher Score:**

The technique doesn't stand out to the level for a spotlight or oral.

**Justification For Why Not Lower Score:**

The paper has a clear and solid contribution and should be accepted.

---

### Decision · Program_Chairs · 2024-01-16

Accept (poster)